# Unmodified rabies mRNA vaccine elicits high cross-neutralizing antibody titers and diverse B cell memory responses

Fredrika Hellgren[1,2,10], Alberto Cagigi[1,2,8,10], Rodrigo Arcoverde Cerveira [1,2,10], Sebastian Ols [1,2], Theresa Kern [1,2], Ang Lin[1,2,9], Bengt Eriksson[3], Michael G. Dodds[4], Edith Jasny[5], Kim Schwendt[5], Conrad Freuling[6], Thomas Müller[6], Martin Corcoran[7], Gunilla B. Karlsson Hedestam[7], Benjamin Petsch[5] & Karin Loré [1,2] ✉

Licensed rabies virus vaccines based on whole inactivated virus are effective in humans. However, there is a lack of detailed investigations of the elicited immune response, and whether responses can be improved using novel vaccine platforms. Here we show that two doses of a lipid nanoparticle-formulated unmodified mRNA vaccine encoding the rabies virus glycoprotein (RABV-G) induces higher levels of RABV-G specific plasmablasts and T cells in blood, and plasma cells in the bone marrow compared to two doses of Rabipur in non-human primates. The mRNA vaccine also generates higher RABV-G binding and neutralizing antibody titers than Rabipur, while the degree of somatic hypermutation and clonal diversity of the response are similar for the two vaccines. The higher overall antibody titers induced by the mRNA vaccine translates into improved cross-neutralization of related lyssavirus strains, suggesting that this platform has potential for the development of a broadly protective vaccine against these viruses.

Rabies virus (RABV) is a neurotropic virus transmitted to humans mainly via bites from infected animals[1]. Once the virus has reached the central nervous system (CNS), acute encephalitis, paralysis and coma precede death. The immune system, therefore, has a limited window to neutralize the virus after transmission before it has entered the CNS. Prevention of mortality and morbidity relies on the effective administration of post-exposure prophylaxis (PEP) promptly after exposure[2,3]. This includes rabies immunoglobulins (RIG) purified from either vaccinated horses or humans in combination with up to five doses of a rabies vaccine. Two or three doses may be administered prophylactically to individuals at risk, e.g. animal shelter staff,

veterinarians or travelers. Around 17 million persons per year are vaccinated and treated after exposure to RABV. The World Health Organization (WHO) currently lists four prequalified rabies vaccines based on inactivated whole rabies virus, one of which is Rabipur, for use as both pre- and post-exposure prophylaxis[3–5]. The Rabipur vaccine has been used for almost 40 years and undoubtably saved numerous lives. However, the need for multiple doses and high costs has led to incomplete protection in endemic areas and about 60,000 deaths per year worldwide[6–8]. It is therefore timely to investigate if modern vaccine technologies can improve the immune responses to vaccination and the prospects of compliance and higher survival rates.

[1]Division of Immunology and Allergy, Department of Medicine Solna, Karolinska Institutet and Karolinska University Hospital, Stockholm, Sweden. [2]Center of Molecular Medicine, Stockholm, Sweden. [3]Astrid Fagraeus Laboratory, Comparative Medicine, Karolinska Institutet, Stockholm, Sweden. [4]Certara USA, Inc, Princeton, NJ, USA. [5]CureVac SE, Tübingen, Germany. [6]Institute for Molecular Virology and Cell Biology, Friedrich-Loeffler-Institut, Greifswald-Insel Riems, Greifswald, Germany. [7]Department of Microbiology and Tumor Biology, Karolinska Institutet, Stockholm, Sweden. [8]Present address: Nykode Therapeutics, Oslo, Norway. [9]Present address: School of Basic Medicine and Clinical Pharmacy, China Pharmaceutical University, Nanjing, China. [10]These authors contributed equally: Fredrika Hellgren, Alberto Cagigi, Rodrigo Arcoverde Cerveira. ✉e-mail: karin.lore@ki.se

There is a reference level of virus neutralizing titers (VNT) suggested by the WHO as the threshold for adequate prophylactic rabies vaccination[9]. The rabies virus glycoprotein (RABV-G) exposed on the surface of the virus is the principal target for neutralizing antibodies and a critical component of a vaccine. RABV-G exhibits five distinct antigenic sites i.e. I, II, III, IV and minor site "a" which all have been identified by utilizing monoclonal antibodies (mAbs) isolated from individuals receiving the current standard vaccination[10–13]. Some of these mAbs (CR57; Rafivirumab and RVC20 binding to Site I as well as CR4098; Foravirumab, 17C7; Rab1 and RVC58 binding to Site III) have been proposed as alternatives to RIG treatment[14–17]. Since several RABV variants are present, mAbs given in combination offer a better and more broadly neutralizing activity[15,18–20]. A desired feature of rabies vaccine candidates is therefore that they generate diverse neutralizing antibodies that target multiple antigenic sites on RABV-G.

A sequence optimized nucleoside-unmodified mRNA vaccine encoding RABV-G complexed in protamine developed over the last decade was shown to induce neutralizing antibodies that protect against lethal rabies virus infection in several animal models[21]. This was the first mRNA vaccine to be assessed in a phase I clinical trial. It was found to be immunogenic with acceptable tolerability profile[22]. Advances in the formulation strategy have since replaced free and protamine-complexed mRNA with mRNA delivered in lipid nanoparticles (LNPs), which resulted in comparable or improved immunogenicity of the RABV-G mRNA vaccine in rodents and non-human primates (NHPs)[23]. The first clinical trial of this formulation tested safety and, among others, seroconversion at low doses (1, 2, and 5 μg) given as one or two doses which was shown to induce neutralizing activity with VNTs above the WHO threshold in 100% of participants in the groups receiving two doses. However, the levels were lower and the kinetics delayed compared to three doses of Rabipur, indicating that improvements in the mRNA vaccination strategy were required to outperform Rabipur[24].

Here, to investigate the quality and characteristics of the immune responses induced by a RABV-G mRNA vaccine compared to Rabipur, we intramuscularly immunized NHPs with a high dose of the mRNA vaccine (100 μg) and characterized the responses in-depth, from the early innate immune activation following priming to the quality and specificities of the antibodies sampled at longitudinal time points up to a year later. The results suggest that the mRNA platform performs as well if not better than the approved vaccine at the dose tested here.

## Results

### Vaccination with mRNA-induced robust and transient innate immune activation

Eighteen Chinese rhesus macaques were divided into three study groups with equal sex and weight distribution (Fig. 1a). Groups 1 and 2 received 100 μg mRNA vaccine, while group 3 received a human dose of the licensed whole inactivated virus vaccine Rabipur. Groups 2 and 3 received a second dose with the respective vaccines at four weeks from the initial dose to compare the characteristics of the responses side-by side using the same immunization strategy. All vaccines were administered intramuscularly. Peripheral blood and bone marrow (BM) aspirates were collected at multiple timepoints during the 50-week study period.

We first investigated differences in the early innate immune activation between the mRNA vaccine and Rabipur. The mRNA vaccine induced significant increases in serum concentrations of interferon alpha (IFNα) as well as IFN-inducible I-TAC/CXCL11, which were not detectable in the Rabipur group (Fig. 1b). In addition, the mRNA vaccine induced higher levels of interleukin 1 receptor antagonist (IL-1RA), monocyte chemoattractant protein 1 (MCP-1), and the chemokine Eotaxin. Cytokines returned to baseline levels by two weeks post-immunization (Fig. 1b). Most tested cytokines showed no or very low systemic increase at 24 h after immunization (Supplementary Fig. 1). In

line with the transient increase in cytokine/chemokine secretion there was a clear increase of circulating CD14 + CD16 + intermediate monocytes at 24 h, which was again more pronounced after mRNA vaccination (Fig. 1c, d). Clinical chemistry and complete blood counts (CBC) demonstrated no or minor fluctuations that remained within the normal range after immunization with either vaccine (Supplementary Fig. 2a, b). Neither vaccine induced increases in body temperature (Supplementary Fig. 3a), nor negatively affected body weight over time (Supplementary Fig. 3b). There were no significant or sustained fluctuations in circulating lymphocyte, total monocyte, and dendritic cell (DC) subsets (Supplementary Fig. 3c). Collectively, we observed a transient innate immune activation which was more pronounced, and type-1 IFN-polarized, with the mRNA vaccine.

### The mRNA vaccine induced higher levels of RABV-G specific antibodies, plasmablasts, plasma cells, memory B cells, and T cells

Two weeks after the prime immunization, all animals except one had developed rabies virus neutralizing antibody (RVNA) titers above the WHO recommended threshold (Fig. 2a and Supplementary Fig. 4a)[4]. In contrast to the mRNA-vaccinated groups, the Rabipur group showed declining titers by four weeks following prime (Fig. 2a). The titers further increased with the boost immunization of both mRNA vaccine and Rabipur. RVNA titers in all three groups were above the WHO-threshold at week 18. Only the boosted mRNA group showed sustained titers above this threshold in all animals beyond week 18 and throughout the 50-week study period (Fig. 2a and Supplementary Fig. 4a). RABV-G binding IgG titers showed similar kinetics to the neutralizing titers (Fig. 2b and Supplementary Fig. 4b). RABV-G-specific IgM levels were detectable in all groups after vaccination (Supplementary Fig. 4c).

Antibody-secreting RABV-G-specific plasmablasts were readily detectable by ELISpot at four days after the boost immunization, with higher frequencies seen in the mRNA prime-boost group compared to the Rabipur group (Fig. 2c). RABV-G specific plasma cells in the bone marrow were detectable after prime immunization in all groups (Fig. 2d). The numbers increased two weeks after boost and showed significantly higher levels in the mRNA vaccinated group compared to the licensed vaccine. While the group that received only one immunization with the mRNA vaccine did not show detectable levels at study end (week 50), RABV-G specific plasma cells remained detectable throughout the study period in both prime-boost groups with no difference in numbers between the groups at this time (Fig. 2D and Supplementary Fig. 4d). RABV-G specific circulating memory B cells (MBC), as measured by binding to fluorescently labeled RABV-G by flow cytometry, showed a clear increase in number after the boost immunization in the mRNA group and remained detectable at 18 weeks after both mRNA and Rabipur prime-boost vaccination (Fig. 2e, f). Importantly, MBC-derived antibody secreting cells could still be detected by ELISpot in the mRNA groups, especially in the prime-boost group, at week 50 (Supplementary Fig. 4e).

RABV-G specific memory T cells, as assessed by antigen recall assay using stimulation with overlapping RABV-G peptides and intracellular cytokine production, showed that the mRNA vaccine groups had low but detectable CD4+ T cell responses after prime immunization with a clear increase after boost (Fig. 2g). The T cell response was Th1 polarized as evidenced by IFN-γ and IL-2 production (Fig. 2h), and there were no or very low cells producing either IL-13, IL-21 or IL-17A (Supplementary Fig. 5a, b). However, it should be noted that intracellular cytokine production is intrinsically more difficult to detect for Th2 and Tfh responses (as evidenced by the SEB controls in Supplementary Fig. 5a), which might therefore lead to overinterpretation of the ratios between Th1 and Th2/Tfh. The Rabipur group showed no detectable CD4+ T cell responses. CD8+ T cell responses were low or undetectable in all groups (Supplementary Fig. 6a–f). Collectively, the

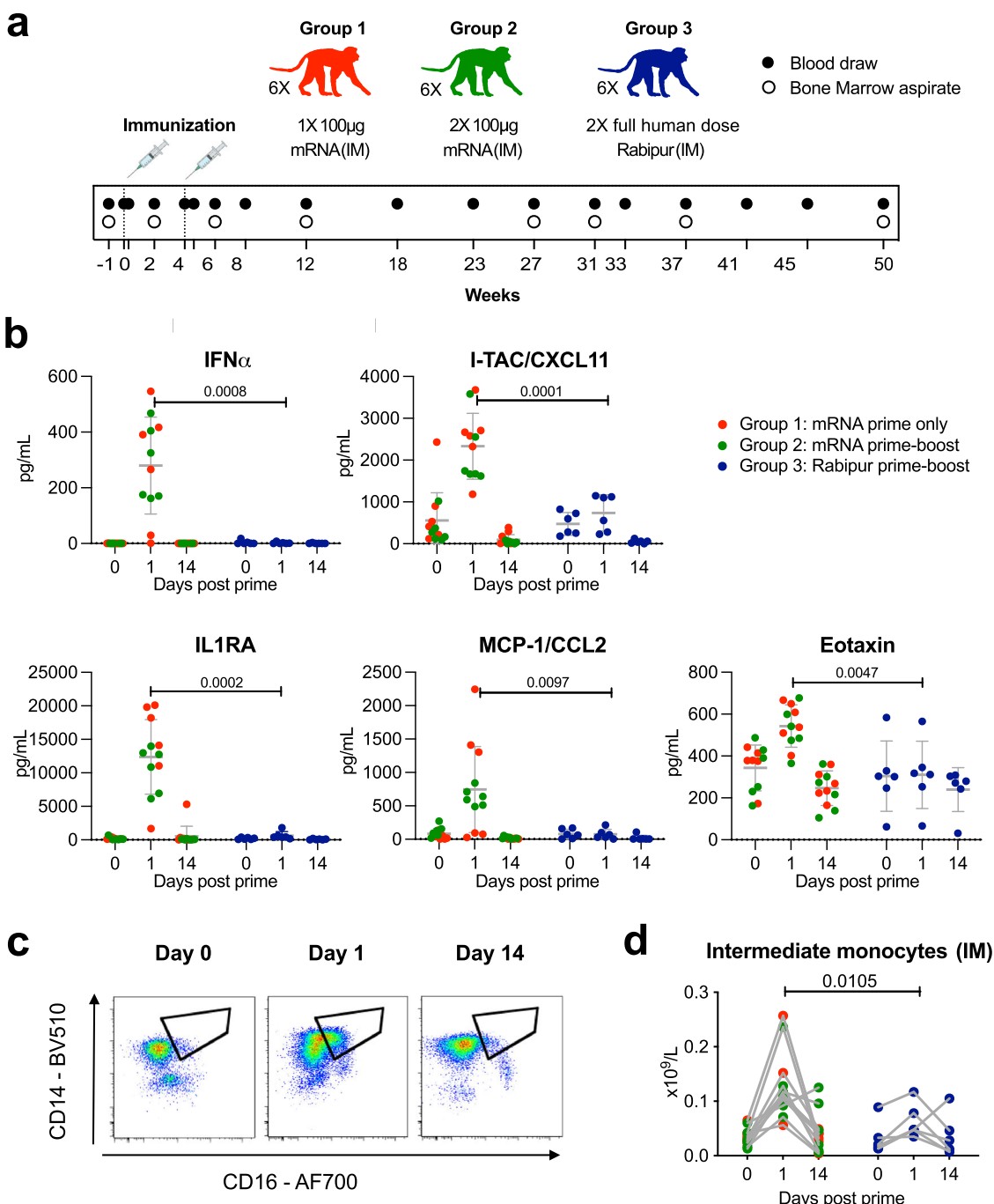

**Fig. 1 | Induction of Type I IFN and intermediate monocytes after mRNA vaccination. a** Outline of study design with respect to group and vaccine assignment, group composition, vaccine dose/route of administration, number of vaccine doses, time interval between each dose and sampling timeline. Red, green and blue will be the color coding for data referring to groups 1, 2 and 3 respectively throughout the manuscript. Full black circles and empty white circles across the study timeline indicate blood draws and bone marrow aspirates respectively. **b** Dot plot of the detectable plasma cytokine and chemokine levels showing significant differences between the groups measured by a 30-plex assay. The values shown refer to the levels measured at day 0 (prior to prime immunization), day 1 (peak response) and day 14 (return to steady-state levels). Direct comparisons are shown for each cytokine between combined groups 1 and 2 (as at this time of the study,

they both received one dose of the mRNA vaccine and are therefore equivalent) and group 3. *n* = 18 biologically independent animals. Statistical differences were assessed at day 1 comparing groups using Mann–Whitney *U* test. Cytokine concentrations undetectable or below lower limit of quantitation (LLOQ) are shown as LLOQ value. Dotted lines indicate LLOQ. Representative flow cytometry plots (**c**) and longitudinal data (**d**) showing the differentiation and temporary fluctuation of intermediate monocytes. Data are shown for day 0 (prior to prime immunization), day 1 (peak innate response) and day 14 (return to steady-state levels). Complete gating strategy for the identification of monocytes is shown in Supplementary fig. 11b. Statistical differences in (**d**) were assessed at day 1 using Mann–Whitney *U* test. *n* = 18 biologically independent animals. All statistical tests comparing the study groups were two-tailed tests. All error bars indicate mean ± SD.

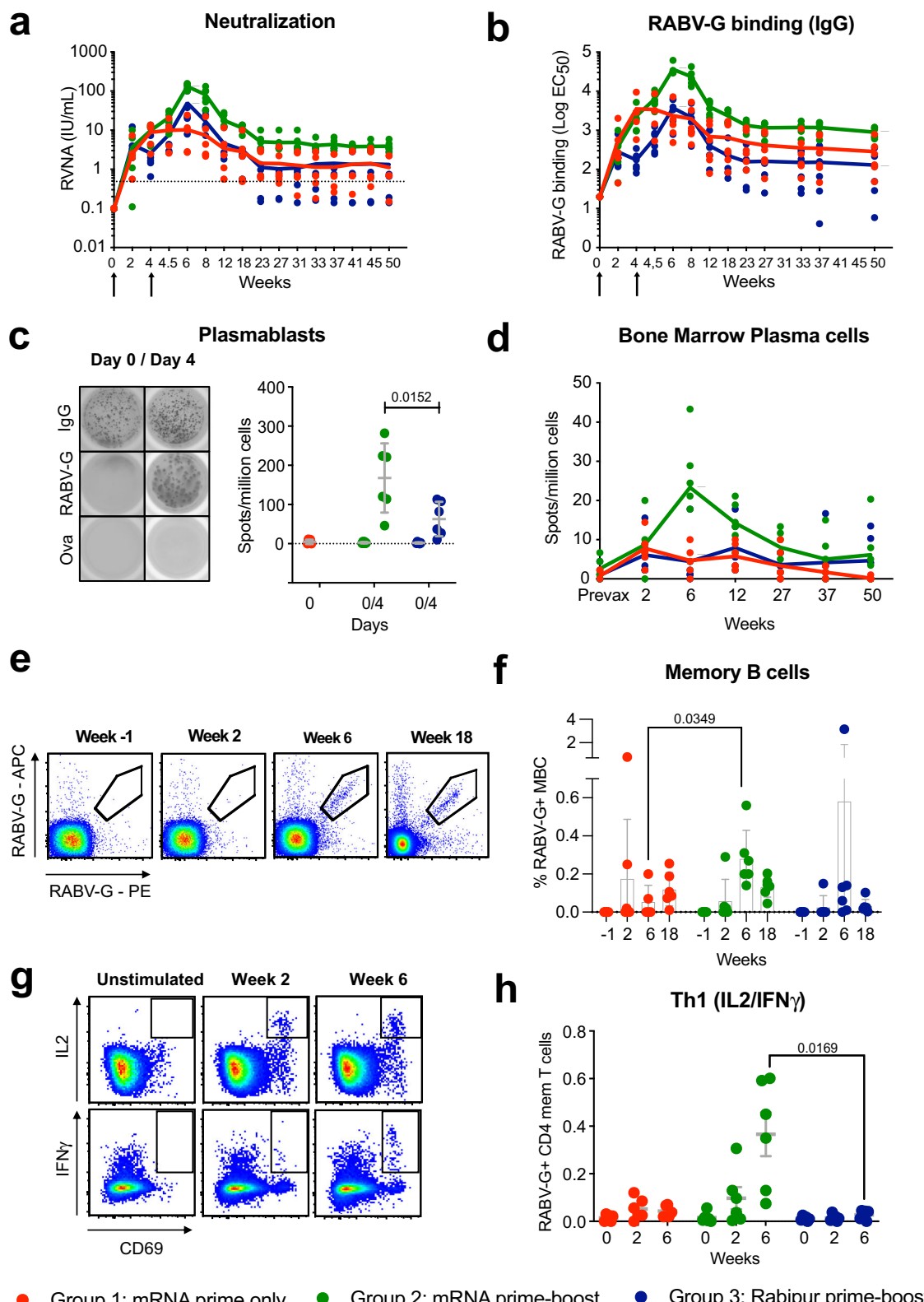

Group 1: mRNA prime only   Group 2: mRNA prime-boost   Group 3: Rabipur prime-boost

data demonstrate that the two-dose strategy with the mRNA vaccine induced significantly higher antibody titers than the inactivated virus vaccine, as well as RABV-G-specific cell populations essential for antibody production and longevity of the response.

There was a strong correlation between RABV-G binding IgG levels and neutralizing titers in all the groups, indicating that the magnitude of these antibodies aligned well with their quality (Fig. 3a). There was

no difference in binding titers to the Rabipur vaccine itself compared to RABV-G in the Rabipur group (Supplementary Fig. 7a) and there was strong correlation between the titers to Rabipur and RABV-G as well as between Rabipur and neutralization (Supplementary Fig. 7b). As Rabipur is an inactivated complete virus vaccine compared to the mRNA vaccine that encodes only RABV-G, it was possible that Rabipur also induced responses to additional antigens. We found that RABV-

**Fig. 2 | Higher level humoral and cellular responses with mRNA vaccination.**
**a** Neutralizing antibody titers are shown across the whole study timeline. Arrows indicate immunizations. Statistical significance was calculated for peak antibody responses (week 6) and at study end (week 50) using Kruskal–Wallis test as per Supplementary Fig. 4a. Connecting lines indicate group means. The dashed line across refers to the RVNA titer of 0.5 IU/mL which is suggested by the WHO as the VNT threshold. $n = 18$ biologically independent animals. **b** Total IgG titers expressed as the half-maximal effective concentration ($EC_{50}$) calculated with a sample dilution series measured by ELISA are shown across the whole study timeline. Arrows indicate immunizations. Connecting lines indicate group means. $n = 18$ biologically independent animals. Statistical significance was calculated for peak antibody responses (week 6) and at study end (week 50) using Kruskal–Wallis test as per Supplementary Fig. 4b. **c** Antigen-specific plasmablasts are measured by B cell ELISpot prior to boost (baseline) and four days after the boost. $n = 18$ biologically independent animals. Statistical differences were assessed at day 4 using Mann–Whitney $U$ test. Error bars indicate mean ± SD. **d** Longitudinal data of antigen-specific antibody-secreting plasma cells in the bone marrow enumerated by B cell ELISpot at the different study timepoints. $n = 18$ biologically independent

animals. Statistical significance was calculated for peak antibody responses (week 6) and at study end (week 50) using Kruskal–Wallis test. Representative flow cytometry plots (**e**) and longitudinal data (**f**) showing the generation and expansion of antigen-specific memory B cells. Data are shown for week-1 (prior to prime immunization), 2 weeks after prime immunization (week 2), 2 weeks after boost (week 6 from study start) and from a later time point (week 18) from which single cells have also been sorted. $n = 18$ biologically independent animals. Statistical significance was calculated at weeks 2, 6, and 18 using Kruskal–Wallis test. Error bars indicate mean ± SD. Representative flow cytometry plots (**g**) and longitudinal data (**h**) showing the generation of antigen-specific Th1 cells identified by their ability to produce intracellular IL-2 and IFN-γ upon stimulation with a pool of overlapping peptides covering the whole RABV-G protein. Data are shown for unstimulated and stimulated cells collected prior to immunization (week 0), 2 weeks after prime (week 2) and 2 weeks after boost (week 6 from study start) immunizations. $n = 18$ biologically independent animals. Statistical significance was calculated at week 6 using Kruskal–Wallis test. All statistical tests comparing the study groups were two-tailed tests. Error bars indicate mean ± SEM.

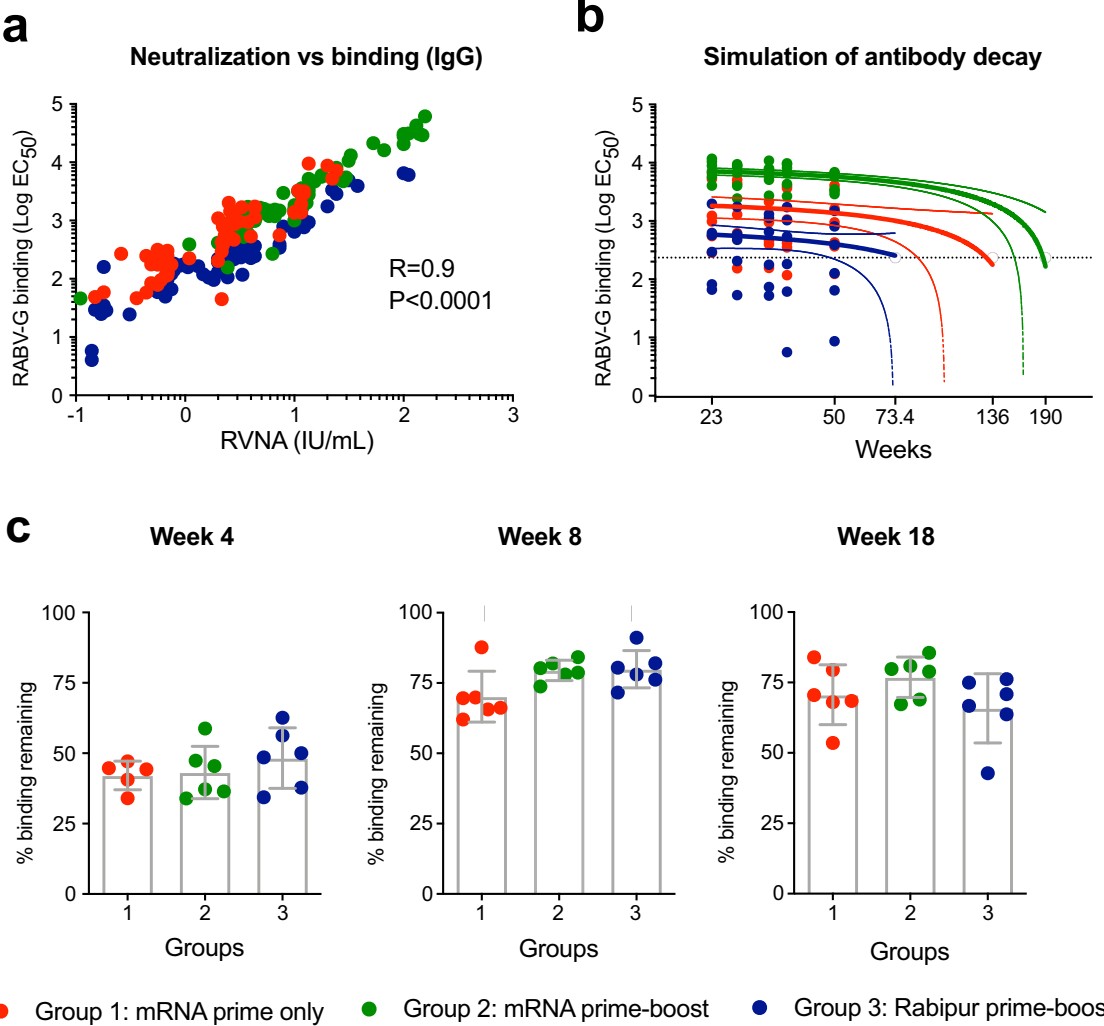

**a** Neutralization vs binding (IgG)

**b** Simulation of antibody decay

**c** Week 4    Week 8    Week 18

● Group 1: mRNA prime only    ● Group 2: mRNA prime-boost    ● Group 3: Rabipur prime-boost

**Fig. 3 | High quality neutralizing antibodies induced by both vaccines.**
**a** Spearman correlation between neutralization (RVNA) and total binding expressed as half-maximal effective concentration ($EC_{50}$) of total IgG binding to RABV-G measured by ELISA. **b** Simulation of antibody decay over time down to the $EC_{50}$ of 80.9 corresponding to 0.5 IU/mL for each group based on antibody half-life estimates. Simulations out to ten years were performed, using the model fits for each animal. The time to binding assay target was calculated for each animal, and

summaries per group are reported. The 95% CI is shown. **c** Antibody avidity by ELISA. The percentage of binding remaining after mild urea wash is shown for plasma collected 1 month after prime immunization (week 4), 1 month after boost (week 8 from study start) and at a later time point (week 18). $n = 18$ biologically independent animals. Statistical significance was calculated using Kruskal–Wallis test. Error bars indicate mean ± SD.

Nucleoprotein (N)-specific IgG levels in the Rabipur group were not significantly increased compared to background binding observed pre-vaccination or the mRNA-immunized group (Supplementary Fig. 7c). This data indicates that the majority of neutralization is mediated by RABV-G-specific antibodies in both groups, as previously described[25]. However, depletion of anti-RABV-G antibodies from the samples would be required to definitively show whether there is also a neutralizing effect of antibodies specific to other RABV proteins.

To address the long-term maintenance of RVNA titers above the WHO-threshold for protection, we calculated half-life estimates for both RVNA and RABV-G binding titers and simulated antibody decay over time using model fits for each animal (Supplementary Fig. 8a, b). Due to the semi-categorical nature of RVNA titers, we performed the simulation with total RABV-G binding titers instead of RVNA and used $EC_{50}$ of 80.9 as a target. This value corresponds to the WHO-threshold of 0.5 IU/mL by data interpolation (Fig. 3b and Supplementary Table 1). The simulation indicated that it may take up to 3.5 years (190 weeks) for RABV-G binding titers induced by prime-boost with the mRNA vaccine to wane below the threshold. With only prime immunization with the mRNA vaccine, it would take 2.5 years (136 weeks) and <1.5 years (73.4 weeks) after prime-boost with Rabipur. However, it should be noted that the animals that received two doses of Rabipur had higher peak neutralization than those that received a single dose of mRNA vaccine (Fig. 2a) despite lower antibody binding titers (Fig. 2b) suggesting a qualitatively better response induced by two doses of whole inactivated virus vaccine.

Together, these data demonstrate that both vaccines generate neutralizing antibodies with high potency, but the mRNA vaccine induced higher titers resulting in more durable responses above the protective threshold. The high quality of antibody responses induced by both vaccines was further supported by the fact that the decrease in binding strength to RABV-G, in the presence of a chaotropic reagent, showed no differences between the animals that received two doses of mRNA vaccine compared to two doses of Rabipur (Fig. 3c).

## The mRNA vaccine and Rabipur stimulated similar levels of somatic hypermutation

Multiple immune parameters demonstrated that the mRNA vaccine induced responses of higher magnitude as compared to licensed Rabipur when comparing an identical two dose, 4-week spaced immunization schedule, which differs from current recommended vaccination schedules for rabies pre-exposure prophylaxis. However, this study design was chosen to enable a qualitative comparison of the elicited antibodies by either vaccine. We therefore performed a more detailed assessment of the groups that had received two doses of vaccine. To assess qualitative differences of the antibodies with regards to clonality and epitope-specificities on RABV-G, we sequenced the immunoglobulin heavy chain variable region (HV) from RABV-G-specific MBC that were single-cell sorted at weeks 8 and 18. We obtained productive and high-quality sequences from 410 and 506 single MBC from the mRNA vaccine and Rabipur prime-boost groups, respectively. The level of somatic hypermutation (SHM) in the HV region was calculated by assigning the sequences to the largest macaque germline IGHV allele database available based on multiple rhesus macaques, KIMDB[26]. Both groups showed a trend toward a continued increase in the per-animal average SHM of RABV-G-specific MBCs, from 2 weeks post boost to 12 weeks post boost. However, a difference in SHM was not observed between the groups (Fig. 4a).

We next assessed whether high SHM is required for rabies virus neutralizing antibody activity. We constructed maximum-likelihood phylogenetic trees from the amino acid heavy chain VDJ sequences from the sorted RABV-G-specific B cells, also including 67 sequences obtained from the animals that received one dose of mRNA vaccine (Fig. 4b). We selected 12 sets of paired heavy and light chain sequences for mAb expression. The selection was made based on the degrees of SHM including four mAbs with the lowest SHM (0–0.7% SHM; named LowRab1-4), four mAbs with medium level SHM (1.7–4.4% SHM; named MedRab1-4) and four mAbs with the highest SHM (5.4–9.2% SHM; named HighRab1-4) (Fig. 4b). Sequence-similarity with therapeutic[14–16] and other previously characterized broad-spectrum reactive rabies virus neutralizing mAbs[17] binding to either Site I or Site III on RABV-G (Fig. 4c) was taken into consideration in the selection. Eight out of the 12 mAbs were found to bind RABV-G (Supplementary Fig. 9a and Fig. 4d). Six of these had binding potencies comparable to the reference rabies virus neutralizing mAbs (Fig. 4d). The characteristics of the eight RABV-G-binding mAbs were further investigated by competition for binding to RABV-G in the presence of the different reference mAbs (Supplementary Fig. 9b). The analysis showed that three mAbs (LowRab1, LowRab2 and HighRab1) strongly competed with Rafivirumab and RVC20 (Site I-binding), and three other mAbs (MedRab1, MedRab3, and LowRab3) moderately competed with either Foravirumab, Rab1, and RVC58 (Site III-binding) (Fig. 4e). HighRab1 showed moderate/low competition with Foravirumab, Rab1, and RVC58 possibly suggesting interference with Site III binding mAbs by steric hindrance and/or because of angled binding to Site I. HighRab2 and HighRab3 did not compete with any of the reference antibodies, although this result may reflect the fact that these two mAbs showed the overall weakest detectable binding out of the expressed mAbs (Fig. 4d). We finally tested the ability of the eight mAbs to neutralize rabies virus and correlated the data with the degree of SHM. Four of the eight mAbs (HighRab1, MedRab1, MedRab3, and LowRab3) showed potent neutralizing activity against rabies virus. RABV-G neutralizing antibodies thus do not require high SHM, but this likely depends on which epitope is targeted (Fig. 4f).

## mRNA vaccination generated broadly reactive antibodies, which were maintained over time

We further assessed the clonality of the response elicited by the two vaccines, and we asked if different epitope specificities account for qualitative differences in the responses. We constructed phylogenetic trees from the RABV-G-specific Sanger sequences (Supplementary Fig. 10a, b). The sequences from known rabies virus-neutralizing mAbs were included as points of reference. The phylogenetic trees from the different groups showed an overall high degree of resemblance, which indicated that both the mRNA and Rabipur vaccine elicited antibodies across most HV gene families (Supplementary Fig. 10a, b). To increase sequencing depth, we performed bulk IgG repertoire sequencing on all animals at week 18, which generated ~12.5 million reads. We thereafter used the heavy chain sequences identified from the single memory B cell sorts as query sequences to identify clonally related sequences in the bulk repertoires. By this method, we identified 987 sequences that were related to the RABV-G-specific Sanger sequences in the IgG repertoire. Both groups induced diverse HV allele usage, consistent with a polyclonal response (Fig. 5a). We observed an enrichment for different alleles among RABV-G-specific B cells which were also frequently used in the total IgG repertoire of our animals, as well as previously described IgM repertoires[27]. There was donor variability in clonal distribution across both groups with some animals having expanded clones while other clones were only represented by a single sequence (Fig. 5b). This suggests that both mRNA and Rabipur vaccination induces mobilization of diverse clones. When we assessed sample coverage per group, the rarefaction curve showed that ~75% of Rabipur sample coverage could be reached when downsampled to 500 sequences, in contrast to only ~50% for the mRNA group. This indicates that the mRNA vaccine might have elicited a more diverse clonotype recruitment and expansion compared to Rabipur (Fig. 5c). The clonotype diversity was further estimated per individual by using a species richness diversity algorithm (Chao1). With this limited sample size, we observed a trend toward higher species richness in the mRNA vaccinated group (Fig. 5d).

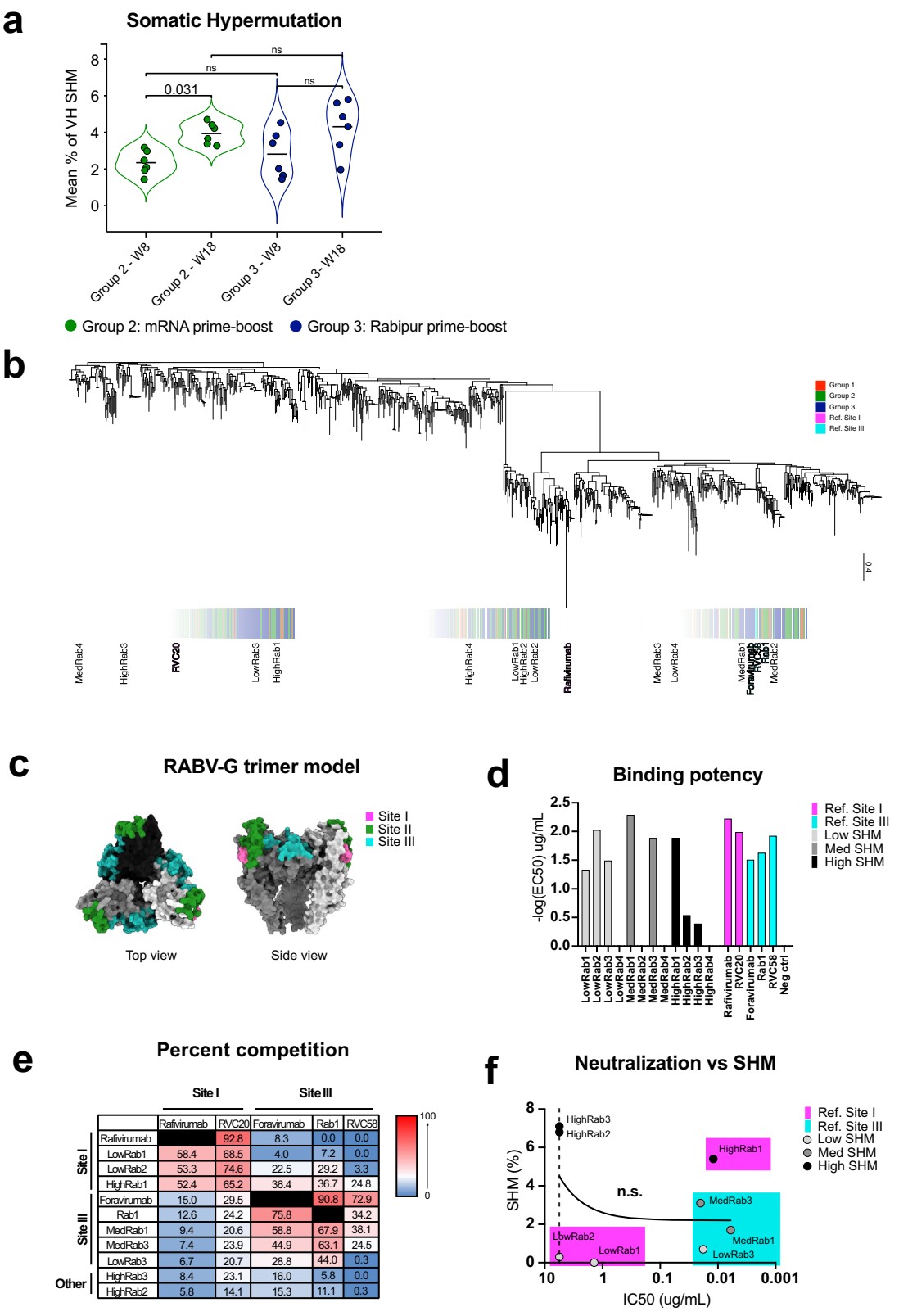

To assess whether there were also diverse binding specificities amongst the circulating antibodies induced by the vaccines, we evaluated competition for binding to RABV-G in the presence of the different reference mAbs in the plasma collected one month after boost. The analysis showed that the plasma antibodies induced in the mRNA vaccinated group targeted all antigenic Sites I, II and III on RABV-G, and had superior competition with all reference mAbs except Rafivirumab

compared to the response induced by Rabipur (Fig. 5e, Supplementary Fig. 12). This demonstrated that a diverse response is elicited by the mRNA vaccine but also by Rabipur although the lower titers in this group may restrict the breadth. Finally, we asked whether the enhanced magnitude and broad antibody response targeting multiple antigenic sites on RABV-G in the mRNA vaccine group could also improve neutralizing activity against lyssaviruses from different

**Fig. 4 | Somatic hypermutation is similarly induced by mRNA and Rabipur and is not required for neutralization. a** Violin plots showing the mean level of SHM of antigen-specific Sanger sequences per animal across different groups and time-points. Lines represent group mean. Statistical significance was calculated using Mann–Whitney test for non-paired data and Wilcoxon signed-rank test for paired data. **b** Maximum likelihood tree inferred by the multiple sequence alignment of VDJH amino acid antigen-specific sequences for all groups. The tree is drawn to scale, with branch lengths in the same units as those of the evolutionary distances used to infer the phylogenetic tree. The scale bar indicates the distance of 0.2 substitutions per sequence position. This analysis involved 334 amino acid sequences. The reference rabies antibodies binding to Site I and to Site III are indicated with magenta and cyan bars respectively. Twelve sequences were selected for cloning monoclonal antibodies based on lowest, medium and highest SHM (named LowRab1-4, MedRab1-4 and HighRab1-4) as well as based on proximity with the reference antibodies when possible. **c** Top and side view of a trimeric RABV-G model with antigenic sites I and III highlighted in magenta and cyan respectively. PMDB ID: PM0079619. **d** Binding potency (EC$_{50}$) of the cloned mAbs is shown together with that of the reference rabies antibodies and negative control. Light gray, dark gray and black bars indicate the mAbs with the lowest, medium and highest SHM (LowRab, MedRab and HighRab) respectively. The magenta and cyan bars refer to the reference rabies antibodies binding to Site I and to Site III respectively. **e** Summary of competition between the cloned mAbs with the reference rabies mAbs. The color gradient from red to blue indicates high to no competition. **f** Spearman correlation between % of SHM and neutralization (IC$_{50}$) in the different cloned mAbs. The dashed line indicates the limit of detection (LOD) for neutralization which corresponds to the highest concentration of mAb used for the assay (5 μg/mL). All statistical tests comparing the study groups were two-tailed tests.

species. Neutralization capacity was tested against rabies virus strain CVS-11 side by side to European bat lyssavirus 1 (EBLV1) and Duvenhage lyssavirus (DUVV) from Phylogroup I, Lagos bat virus (LBV) of Phylogroup II and Lleida Bat Lyssavirus (LLBEV) of Phylogroup III. Cross-neutralization capacity was tested in samples taken at 47 weeks post prime immunization when antibody levels in all animals had waned to a plateau phase. All the mRNA vaccinated animals showed cross-neutralizing responses to EBLV1 and DUVV, while the neutralization was significantly lower in the Rabipur immunized animals (Fig. 5f). No neutralization to LBV or LLBEV was found in either group. In conclusion, our data demonstrate that a RABV-G encoding mRNA vaccine can induce higher levels of cross-neutralizing antibodies compared to Rabipur.

## Discussion

Relying on affordable vaccines, the WHO targets zero deaths from rabies by 2030[28]. The currently licensed rabies vaccines that are based on whole inactivated virus have demonstrated acceptable efficacy and tolerability. However, they are amongst the most expensive vaccines on the market and require from two to five doses depending on the protocol followed and whether the vaccine is administered pre- or post-exposure[6,29,30]. The high number of required doses often results in incomplete compliance with the full vaccination regimen in endemic areas, mainly in Asia and Africa[6–8,28,31,32]. The mRNA vaccine platform has the potential to overcome some of the challenges associated with vaccine cost and dose requirements. Cell-free in vitro transcription of mRNA could be made more cost-effective than purification of cell-culture grown infectious virus, and it is scalable as made evident during the SARS-CoV-2 pandemic[33]. Nucleoside-unmodified mRNA vaccines encoding RABV-G were tested in clinical phase I studies with the goal to develop a vaccine requiring fewer doses[22,24]. However, further development is needed to identify a dose with acceptable tolerability while being highly immunogenic[23,24]. The primary aim of this study was to characterize the magnitude and quality of the immune responses induced by an RABV-G mRNA vaccine compared to those elicited by the licensed whole inactivated virus vaccine, Rabipur, after intramuscular injection. Importantly, the study was performed using the same dosing schedule to allow for direct comparison between the vaccines. Further studies are therefore needed to benchmark against Rabipur given at recommended schedules using up to five doses.

The characteristic induction of type I IFN and innate immune activation likely plays a role in the induction of Th1 polarized antigen-specific adaptive responses following mRNA vaccination[34–37]. Type I IFN responses have been shown to directly support B cell differentiation and survival, resulting in enhanced antibody responses[38,39]. Type I IFN-inducing adjuvants such as TLR3, 7/8 and 9-ligands (poly IC:LC, R848, CpG) result in increased antibody half-life and durability of humoral responses[40–44]. In the current study, we observed that the mRNA vaccine induced a strong and transient type I IFN response

(IFNα, CXCL11), as well as monocyte activation evidenced by MCP-1 secretion and increased levels of intermediate monocyte differentiation within 24 h of administration. We and others have previously found a similar activation profile after administration of nucleoside-modified and unmodified mRNA vaccines[23,35–37,45]. Intermediate monocytes are important for antigen presentation to CD4+ T cells[46] and to support the differentiation of naïve B cells into antibody-secreting plasmablasts[47]. The innate immune profile induced by the mRNA vaccine may therefore contribute to the generation of higher levels of antigen-specific Th1-skewed CD4+ T cells, plasmablasts, and memory B cells compared to whole inactivated virus formulations such as Rabipur.

Prevention of rabies after virus exposure relies on prompt antibody-mediated neutralization before RABV enters the CNS[1]. In our study, one immunization with the mRNA vaccine induced antibody titers comparable to two doses of Rabipur. However, neither of these study groups consistently maintained long term RVNA titers above the WHO recommended threshold. In contrast, all animals in the mRNA prime-boost group developed consistently higher titers compared to Rabipur and presented RVNA titers well above the WHO threshold one year after the first immunization. Thus, the mRNA vaccine may be well suited for pre-exposure prophylaxis with a two-dose regimen. Meanwhile, complete seroconversion and stable immunity in humans have been documented only after administration of at least three doses of Rabipur[48,49]. The mRNA vaccine may also represent an attractive post-exposure strategy, especially in situations when compliance to the multidose vaccination regimen with the current vaccines and availability of RIG/mAbs is poor[7,8]. The ability of mRNA vaccines to induce long-lived immune responses may also present a useful vaccination strategy against other pathogens for which whole inactivated vaccine formulations are currently used. Waning of vaccine-induced antibody titers can increase the risk of breakthrough infections, as shown for mRNA vaccines against SARS-CoV-2[50,51]. Such data highlight the importance of monitoring antibody durability in vaccination studies. In this study, we found that the mRNA vaccine platform induced antibody titers of similar durability to the benchmark whole inactivated virus vaccine.

Although the mRNA vaccine showed higher levels of cellular responses and antibody levels, both vaccines induced antibodies with high neutralizing capacity and avidity (binding strength). In fact, correlation of binding titers to RVNA titers indicated slightly higher neutralizing potency, or ratio of neutralization to total binding, in the Rabipur group compared to mRNA (Fig. 3a). Thereby, the improved neutralization observed with the mRNA vaccine is likely mainly explained by increased antibody titers. To understand whether qualitative differences were present in the memory B cell compartment, we sequenced the immunoglobulin heavy chain variable region of RABV-G-specific memory B cells and assessed the degree of SHM by assignment to an established database of rhesus macaque germline sequences[26]. We observed similar levels of SHM after two doses of

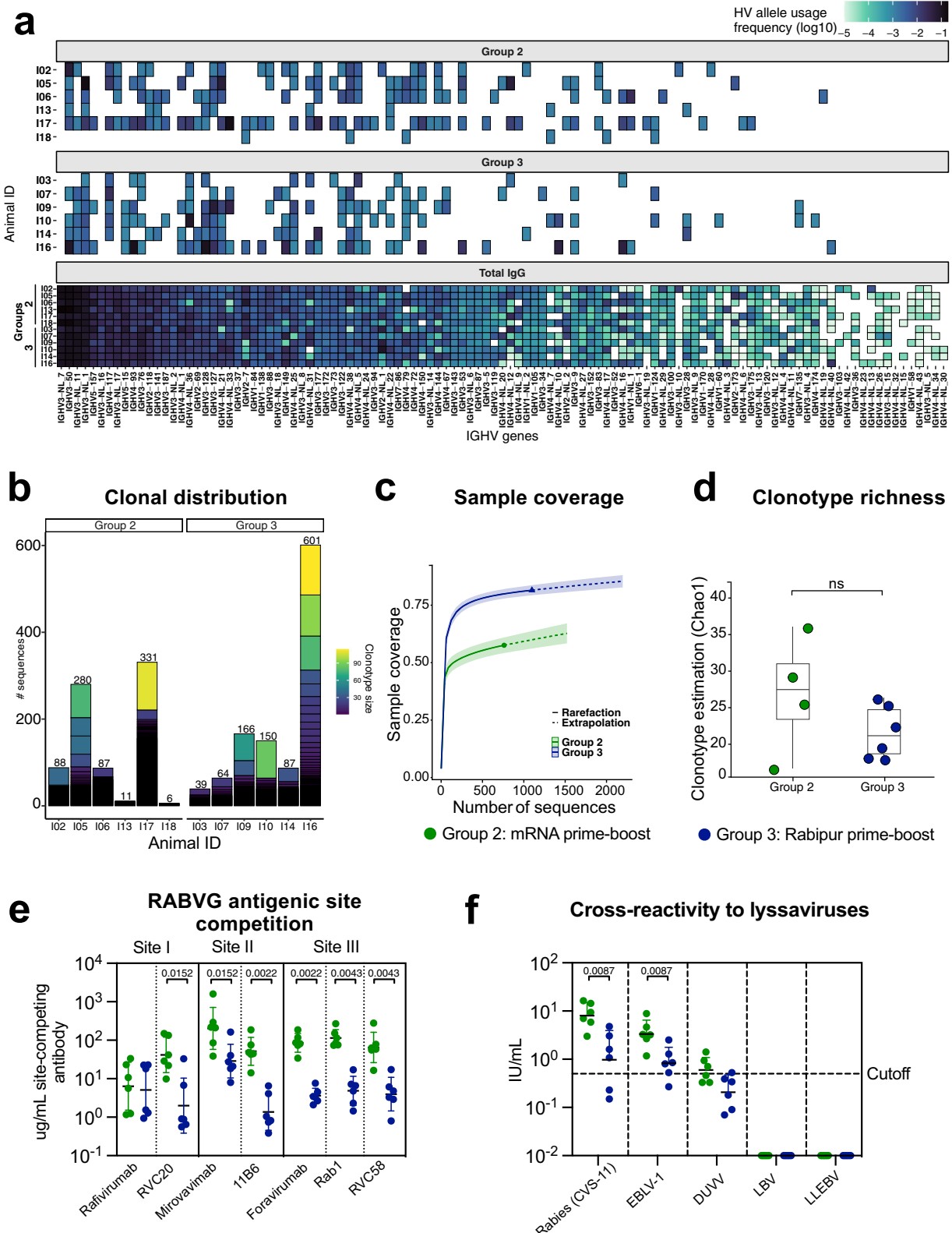

**a**

**b** Clonal distribution

**c** Sample coverage

**d** Clonotype richness

**e** RABVG antigenic site competition

**f** Cross-reactivity to lyssaviruses

● Group 2: mRNA prime-boost   ● Group 3: Rabipur prime-boost

either mRNA vaccine or Rabipur. We and others have shown that high levels of germinal center (GC) responses and Tfh activation are induced by mRNA vaccines encoding various viral antigens[36,52]. A high degree of SHM often conveys improved antibody binding affinity to the antigen, which has been shown to be critical for neutralization of several viral pathogens including HIV-1, influenza virus, and the Hepatitis C virus (HCV)[53,54]. While we describe examples of mAbs with

very low SHM that were potent rabies neutralizers, as also observed for other acute cytopathic viruses such as the rabies-like vesicular stomatitis virus (VSV)[55,56], the role of SHM for a larger set of rabies neutralizing antibodies remains to be investigated.

Using high-throughput sequencing (HTS) of bulk IgG repertoires from the vaccinated animals, Sanger sequencing of isolated RABV-G specific B cells and a set of expressed mAbs, we demonstrated that

**Fig. 5 | mRNA induces higher levels of diverse and cross-neutralizing antibody compared to Rabipur. a** Heatmap showing IGHV allele usage for each vaccinated group per animal including both Sanger sequences and bulk HTS. Total IgG repertoires were subsampled to 100,000 sequences per individual, 1.2 million sequences are shown. The scale was adjusted for the total number of sequences for each group, the resulting value was log10 transformed. **b** Bar graph showing the proportion of different clusters of clonally related sequences in the merged datasets (defined as those with the same V and J allele, exact HCDR3 length match, at least one identical nucleotide junction, and 80% amino acid identity of HCDR3). **c** Sample coverage estimation based on clonotype counts for each group, where a value of 1 represents that the entire population was sampled. The continuous colored lines represent the mean, while the surrounding color for each line represents a 95% confidence interval. **d** Species richness estimation (Chao1) for clonotype counts for both groups per individual. Estimation was subsampled 100x to the lowest number of sequences, individual mean values were plotted. The

boxplot centre measurement is the mean, the bounds of box represent the interquartile range (50% percentile), the whiskers represent 1.5 times upper or lower interquartile range. $n = 10$ biologically independent animals. Statistical significance was calculated using Mann–Whitney test, $p$-value = 0.48. **e** Summary of plasma competition between circulating antibodies with seven reference rabies mAbs, quantified as ug/mL site-competing antibody per plasma sample. Plasma competition was measured at week 8 (two weeks post boost). $n = 18$ biologically independent animals. Error bars indicate geometric mean ± geometric SD. **f** Serum neutralizing titers against several lyssavirus strains measured at week 47 post study start (42 weeks post boost). Data is shown as WHO International Units. 0.5 IU/mL was considered as the cutoff for positive neutralization and is displayed as a dotted line. $n = 18$ biologically independent animals. Statistical significance was calculated using Mann–Whitney test. All statistical tests comparing the study groups were two-tailed tests. Error bars indicate geometric mean ± geometric SD.

both vaccines induced diverse memory B cell repertoires. The Rabipur group had a lower number of unique clones despite slightly higher sequencing depth. However, more data would be needed to investigate whether there are differences in inducing diverse B cell clones. In any event, our data show that the mRNA vaccine does not appear to restrict the response compared to Rabipur. We confirmed that the expressed mAbs were specific for different epitopes on RABV-G and that some were potent neutralizing antibodies. The ability of the mRNA vaccine to elicit antibody responses with broad specificity on RABV-G was also found in the plasma from vaccinated animals. It is conceivable that the chemical inactivation of virus with β-propiolactone as used for Rabipur can alter antigen properties and potentially destroy certain epitopes as found for other virus preparations[57,58]. In addition, it has been demonstrated in a mouse model that membrane-bound antigen from mRNA vaccination was more potent than a protein vaccine at activating B cells by lowering the affinity threshold needed for recruitment of cells into GCs[59]. A broad response covering more antigenic sites on RABV-G can be important for neutralizing activity against different RABV variants and lyssaviruses. We noted enhanced cross-neutralization in the mRNA vaccine group in our study, although this may primarily be due to the higher titers induced but nonetheless an important effect with this vaccine.

Nucleic acid vaccines including DNA vaccines against rabies have been successfully tested in animal models for many years[21,23,60]. The mRNA vaccine in the current study has been developed over several years. The first-generation mRNA vaccine encoding RABV-G was complexed with protamine and showed promising results both in terms of safety and immunogenicity in humans up to 400 μg vaccine either intramuscularly or intradermally by needle-free device[22]. A similar mRNA vaccine formulated with LNP yielded unacceptably high reactogenicity after intramuscular administration of 5 μg[24]; however, 1 or 2 μg doses were well tolerated. Prime-boost regimens with these lower doses elicited RVNA titers above the WHO-threshold, but titers were lower and with slower kinetics than those elicited with standard three doses of Rabipur[24]. In our study, administration of two 100 μg doses of the same LNP formulated mRNA vaccine in NHPs did not show any safety concerns with regards to body temperature, hematology, or toxicity. In addition, two doses of the mRNA vaccine showed improved immunogenicity compared to two doses of Rabipur. However, NHPs may be generally more resistant to vaccine adverse events compared to humans, and this caveat must be considered when interpreting safety data. Differences in mRNA vaccine reactogenicity between humans and NHPs may be explained by biological differences in tolerance to type I IFN-associated inflammation[61]. Modifications of the mRNA may also be needed to alter type I IFN while maintaining good antigen expression and high-quality antibodies. Moreover, for this mRNA platform, improvements in the non-translated regions of the mRNA have been described and demonstrated to be more immunogenic and protective in an NHP model of SARS-CoV-2[62]. Further studies

on vaccine dosing are required to strike a balance between tolerable innate immune activity, the magnitude and quality of adaptive immune responses, and the durability of immune protection provided by unmodified mRNA vaccines.

In summary, here we show that immunization of NHPs with two doses of 100 μg of nucleoside-unmodified LNP-formulated mRNA encoding RABV-G induced more potent and durable RABV neutralization as compared with a two-dose regimen of the licensed vaccine Rabipur. Two doses of the mRNA vaccine generated RVNA titers predicted to remain above the WHO threshold for more than three years. Since stronger neutralization and higher frequencies of B cells and plasma cells were induced by the mRNA vaccine, this platform could be an advantageous alternative to the currently licensed rabies vaccines both for pre- and post-exposure prophylaxis. Also, an mRNA vaccine may circumvent issues with production and purification of whole inactivated virus vaccines and can be designed to express RABV-G with optimal conformation and as a stabilized prefusion trimer[63]. Studies to further refine mRNA design, dosage, and immunization regimens will need to be conducted in humans. Nonetheless, the current study provides important mechanistic insight into the responses induced by mRNA vaccines at large, which helps in the development of this type of vaccine modality for RABV and other pathogens.

## Methods
### Vaccines
An experimental and a licensed vaccine against rabies were used in this study. These were a nucleoside-unmodified mRNA encoding for the rabies virus glycoprotein (RABV-G) of the Pasteur strain (GenBank accession number: AAA47218.1) produced by CureVac SE and encapsulated using the lipid nanoparticle (LNP) technology of Acuitas Therapeutics (Vancouver, Canada) as previously described[23], and the inactivated whole rabies virus vaccine Rabipur (of the FLURY LEP strain) that was used as a benchmark control. For simplicity, the experimental vaccine is referred to as "mRNA" throughout the manuscript. Each vaccine dose consisted of either 100 μg of mRNA or the whole recommended dose of Rabipur for human use, which were both prepared in 0.5 mL of sterile 0.9% NaCl.

### Study design, sample collection and assessment of vaccine safety
This study (18427–2019) was approved by the Stockholm Regional Ethical Board on Animal Experiments. Eighteen Chinese rhesus macaques (*Macaca mulatta*), nine males and nine females, were used in this study. The animals were kept in accordance with applicable guidelines and the work was carried out in accordance with European guidelines for animal welfare and care according to the Federation of European Laboratory Animal Science Associations (FELASA). Males and females were housed separately in two large cages at the Astrid Fagraeus

Laboratory (AFL) at the Karolinska Institutet. Animals were divided into three study groups keeping similar distribution of sex and body weight. Group 1 and group 2 received the mRNA vaccine whereas group 3 received Rabipur. Groups 2 and 3 received a boost immunization after 4 weeks from prime whereas group 1 received only the prime immunization. All vaccines were administered by intramuscular (IM) needle injection in the right deltoid. Peripheral blood and bone marrow (BM) aspirates[64] were collected starting at 1 week prior to the prime immunization and at different timepoints for the following 50 weeks. A summary of the study design is shown in Fig. 1. Health parameters such as body weight and temperature were monitored for all animals prior to each sampling while additional safety data, i.e., clinical chemistry (primarily to monitor liver and kidney function) and complete blood counts (CBC), were collected at the day of immunization, after 1 day and after 14 days. Clinical chemistry and complete blood counts were performed by Adlego Biomedical. Approximate T cell and B cell counts were calculated from the respective cell subset proportions obtained from flow cytometry data (described later), and the total lymphocyte count obtained from complete blood counts.

## Quantitation of plasma cytokines and chemokines related to innate immune activation

The modulation of a set of plasma cytokines and chemokines related to innate immune activation during the early phases of vaccination was analyzed using the Procartaplex™ Cyto- and Chemokine NHP 30-plex panel kit (ThermoFisher) according to manufacturer's instructions. Samples were acquired on a Luminex xMAP instrument interfaced to the Bioplex Manager Software (Bio-Rad). The absolute cytokine values obtained were interpolated to a standard curve to calculate the final concentration.

## Assessment of monocyte differentiation after vaccination

The differentiation of monocytes and other leucocyte subtypes including neutrophils, dendritic cells (DC), natural killer (NK), T and B cells was assessed on freshly isolated peripheral blood mononuclear cells (PBMC) collected the day of prime immunization and then after 1 day and after 14 days, by flow cytometry (Supplementary Fig. 11a). Different combinations of fluorescently labeled antibodies against different lineage markers were used (Supplementary Table 2a) and identified as shown in Supplementary Fig. 11c, d.

## Assessment of antigen-specific T cell responses after vaccination

Antigen-specific memory T cells were analyzed in batches from PBMC cryopreserved at different timepoints based on intracellular cytokine production upon antigen recall as previously described[40]. Briefly, 1–2 million PBMCs were stimulated with 1 μg/ml overlapping peptides spanning the RABV-G protein (JPT) in the presence of 10 μg/mL Brefeldin A (Sigma-Aldrich), with 1 μg/mL SEB as positive control and media with DMSO as negative control. Samples were acquired on a BD LSRFortessa flow cytometer and data was analyzed using Flowjo version 9 (BD Life Sciences). After stimulation, cells were stained with a panel of fluorescently labeled antibodies (Supplementary Table 2d), and acquired using a BD LSRFortessa flow cytometer (Becton Dickinson). Data was analyzed using Flowjo version 10. T cells producing cytokines in response to antigen stimulation were identified as shown in Supplementary Fig. 5a.

## Assessment of rabies virus neutralization

Longitudinal assessment of rabies virus neutralizing antibodies (RVNA) was performed by rapid fluorescent focus inhibition test (RFFIT) using the Challenge virus standard strain (CVS-11), using plasma samples from all timepoints, and expressed as international units (IU/mL) according to the standard procedure[65]. Assessment of cross-neutralization to diverse lyssaviruses at week 47. Assessment of RABV-specific neutralizing antibody (VNA) was performed using a

modified fluorescence focus inhibition test (RFFIT), essentially as described previously using RABV (CVS-11) as the test virus[66]. Sera were tested in duplicate in twofold serial dilutions on BHK21-BSR/5 (CCLV-RIE 0194/260) cells with a starting dilution of 1:10. For lyssavirus cross-neutralization, the test virus was replaced with representatives of phylogroup I, II and III lyssaviruses by following the same approach as described previously[67,68]. Specifically, sera were tested for the presence of cross-neutralizing VNA against EBLV-1 (phylogroup I; FLI-ID 7467; GenBank: DQ522866.1), DUVV (phylogroup I; FLI-ID 12863; GenBank: EU293119), LBV (phylogroup II; FLI-ID 12859; GenBank: LN849915.1) and LLEBV (phylogroup III, FLI-ID 40299, GenBank: NC_031955.1). To avoid cell adaptation, test viruses were propagated on mouse neuroblastoma cells (NA 42/13) and passaged no more than three times. For phylogroup I lyssaviruses, the calibrated WHO international standard immunoglobulin (2nd human rabies immunoglobulin preparation, National Institute for Standards and Control, Potters Bar, UK) adjusted to 0.5 (CVS-11), 1.5 (EBLV-1), and 5.0 (DUVV) international units (IU) served as positive control, whilst this positive standard is not neutralized by phylogroup II and III lyssaviruses[67,68]. In all assays, a naïve serum was used as a negative control. Endpoint VNA titres were calculated by fitting a sigmoidal function using non-linear regression as implemented in R studio (R Core Team R 2020) and subsequently converted into concentrations expressed in IU/ml using a value of 0.5 IU/ml as cut-off for seropositivity.

## ELISA for RABV-G binding IgG and IgM

Total binding against RABV-G was assessed by enzyme-linked immunosorbent assay (ELISA). Briefly, 96-well plates were coated with 100 ng/well of recombinant RABV-G (Proteogenix, custom synthesis in HEK293 expression system) of the Pasteur strain and incubated with a duplicate 8-point serial dilution of each plasma sample, in blocking buffer consisting of 5% dry milk diluted in PBS. Detection was performed with goat anti-monkey IgG or IgM HRP-conjugated secondary antibodies (cat# 246-GAMon/IgM(Fc)/PO and cat# 246-GAMon/IgG(Fc)/PO, Nordic MUBio) followed by incubation with TMB substrate (BioLegend; cat# 421101) and stopped with a 1 M solution of $H_2SO_4$. Blocking and washing steps were used when appropriate. Absorbance was read at 450 nm and 550 nm background correction using an ELISA reader. For binding to Rabipur, the same procedures were used with an additional step of pre-coating of the ELISA plates with Lectin from Galanthus nivalis (Sigma) to allow coating of Rabipur components onto the plate. For IgG, the half-maximal effective concentration ($EC_{50}$) was calculated for each sample based on the whole dilution series using Graphpad Prism version 8. For IgM, raw OD values at 1:4 dilution was used as readout. The same procedures were used to assess the binding of the cloned mAbs.

## ELISA for N protein binding IgG

Total binding against N protein was assessed by ELISA as described above for RABV-G. 96-well plates were coated with 100 ng/well recombinant N protein (Proteogenix, Uniprot ID: Q0PNB8). Plates were blocked for 1 h at RT using 5% dry milk in PBS and 7-point serial dilutions of plasma samples added to plates and incubated a further 2 h at RT. Secondary antibody and ELISA development were performed as for RABV-G ELISAs above. Plates were read at 450 nm with 570 nm background correction using an ELISA reader. $EC_{50}$ values were calculated using 4PL curve fits performed in Graphpad Prism version 9.

## Antibody half-life estimation and simulation of antibody decay over time

Data lower than the maximum concentration value (Cmax) for each animal were censored, and time-after-Cmax (TAM=Week-Tmax) was computed. A two-compartment model was fit to TAM and concentration data. The terminal half-life is derived from these model estimates. Simulation of antibody decay over time was performed using R based

on antibody half-life estimates down to the EC$_{50}$ of 80.9 which corresponded to 0.5 IU/mL by data interpolation, for each group.

### Detection of antigen-specific antibody secreting cells in blood and bone marrow

The presence of antigen-specific antibody secreting antigen-specific plasmablasts in blood and plasma cells in the bone marrow were measured directly by Enzyme-linked Immunospot (ELISpot) assay while the presence of antigen-specific memory B cells in blood included a prior additional step of a four-day polyclonal activation of memory B cells in vitro[36,40]. The generation of plasma cells and memory B cells was assessed using frozen BM and PBMC samples while freshly isolated PBMCs were used to assess plasmablasts. Briefly, 96 well ELISpot plates (Merck/Millipore) were coated with 500 ng/well either purified RABV-G, anti-IgG (Jackson ImmunoResearch) (to detect total IgG as a positive control), and ovalbumin (Invivogen) (as a negative control). Cells were then resuspended in R10 complete media (RPMI (Gibco) with 10% FBS, 1% penicillin/streptomycin and 1% L-glutamine), loaded into the plates and incubated overnight at 37 °C. Plates were then washed with phosphate-buffered saline containing 0.05% Tween-20 (PBST). Detection was performed using 12.5 ng/mL of biotinylated anti-human IgG (Jackson ImmunoResearch). After another wash with PBS-T, plates were then incubated with streptavidin conjugated to alkaline phosphatase (Mabtech) diluted 1:1000 in PBS-T. Plates were finally developed using BCIP/NBT (Mabtech). Spots were counted with an AID ELISpot reader and version 6.0 of the associated software (Autoimmun Diagnostika). Background subtraction was performed using OVA-specific spot counts/million plated cells. The calculated background-subtracted number of spots per million cells was used as readout.

### Antibody avidity

The ability of plasma antibodies to retain binding to RABV-G in the presence of a chaotropic reagent was tested by ELISA for plasma collected at weeks 4, 8, and 18. Briefly, for each sample, the dilution factor providing a fixed OD was calculated by interpolation of the previously collected data from the plasma dilution series described above. Such dilutions were then tested in duplicate against RABV-G with or without incubation with 4.8 M urea. The target OD values were selected based on the shared maximal absorbance before binding saturation separately for each timepoint. These were OD = 1.3 for samples taken at week 4 and OD = 1.8 for samples taken at weeks 8 and 18. Antibody avidity was then expressed as % of binding remaining in the urea-treated samples as compared with the non-treated.

### Quantitation and sorting of single antigen-specific memory B cells by flow cytometry

PBMC isolated at different timepoints were thawed in batches and stained with a panel of fluorescently labeled antibodies to identify memory B cells (Supplementary Table 2b, c and Supplementary Fig. 11a, b). Within this population, rabies-specific cells were then additionally identified with a RABV-G probe which was prepared in house. Briefly, recombinant RABV-G protein was biotinylated using the ThermoFisher FluoReporter Mini-Biotin-XX Protein Labelling kit according to manufacturer instructions and then conjugated to streptavidin-fluorochrome complexes. For improved accuracy, cells were stained with RABV-G labeled with 2 different fluorochromes. All samples, with the exception of week 18, were acquired on a BD LSRFortessa flow cytometer. Week 18 samples were acquired on a BD FACSAria Fusion in order to combine data acquisition with simultaneous single-cell sorting into 96-well PCR plates. Data was analyzed using Flowjo version 10 (Treestar). Background subtraction was done using the percentage of RABV-G probe-positive events observed in pre-vaccination samples; since all animals were rabies-naïve, any positive

signal in these samples was considered to represent background fluorescence. The background-subtracted percentage of antigen-specific memory B cells of total IgG-positive memory B cells was used as readout.

### Amplification of VDJ/VJ genes by single-cell PCR, sequencing and cloning of monoclonal antibodies (mAbs)

RNA from single RABV-G-specific sorted memory B cells was extracted and reverse-transcribed into cDNA with random hexamers using the Superscript III kit (Invitrogen) following manufacturer's instructions. Single-cell nested PCR was then performed by using primers and procedures previously reported by Sundling et al. [69]. After single-pass sequencing of the second-round PCR products, the germline V and J allele usage of each heavy chain was determined by assignment to KIMDB[26], while germline V and J allele usage of the light chains was determined by assignment to the IMGT macaque database. Cloning of the final PCR products into human Igγ1, Igκ, or Igλ expression vectors[26] and production of mAbs was then outsourced to GenScript. Clonally related sequences were defined using IgDiscover as those with the same V and J alleles, identical HCDR3 length, minimal 80% aa identity of HCDR3, and at least one junction without a mismatch.

### Bulk BCR library preparation and repertoire sequencing

The library for high-throughput sequencing by Illumina MiSeq 2 × 300 bp paired-end sequencing was based on a 5-prime multiplex PCR method, as previously described[26]. IgG library preparation was done at week 18 post-study start. The RNA extraction was performed using RNeasy Mini Kit (Qiagen) following the manufacturer's instructions. The cDNA synthesis, multiplex PCR, and Index PCR steps were performed following the previously described protocol[26]. The library pool and final sample preparation for the sequencing were performed using MiSeq Reagent Kit v3, 600 cycles (Illumina) following the manufacturer's instructions.

### Evolutionary relationships of taxa and B cell repertoire analyses

Raw sequencing data from Sanger sequencing was filtered for quality using the R/Bioconductor package scifer (v 1.0.0 https://doi.org/10.18129/B9.bioc.scifer) using default settings. Good quality sequences were aligned using IgDiscover[70] (v 0.15.1), as described above, and the full VDJ aa sequences were used for phylogenetic comparisons. Sequences were first aligned using MUSCLE (v 5.1)[71], then the aligned sequences were used for generating maximum-likelihood phylogenetic trees with the LG amino acid evolution model using FastTree (2.1.11)[72,73]. Trees were plotted using the ggtree R package[74].

### IgG bulk repertoire sequencing and lineage tracing

Reads processing was performed using the IgDiscover program[70] (v. 0.15.1) with default settings, except for UMI length of 21 nucleotides and 0 iterations., and using the KIMDB, version 1.1 as the reference database. To integrate the antigen-specific Sanger sequences with the bulk repertoire, we used the antigen-specific sequences to query the bulk IgG repertoire sequencing to trace their clonal relatives. To achieve that, we used the IgDiscover *clonoquery* and *clonotype* modules[75]. We first identified the clonotypes present in antigen-specific sequences for each individual using the clonotype function with the same clonotype definition as mentioned previously. Subsequently, we used the IgDiscover *clonotype* module outputs containing one representative sequence for each clonotype for every individual to query their clonal relatives in their respective bulk IgG HTS dataset using the *clonoquery* function. The summary of sequencing information per animal, such as Sanger sequence counts, raw reads, filtered good quality sequences, number of clonotypes, and number of related sequences found in bulk are shown in the supplementary material (Supplementary Table 3).

## Clonotype diversity estimation

The estimation of sample coverage interpolation and extrapolation was calculated using the iNEXT R package based on the clonal sizes of the merged datasets[76]. The species richness estimation was calculated using the Chao1 algorithm with the iNEXT R package. The estimation was performed at the group level and per individual. For group clonotype richness estimation, we merged all sequences per group and calculated the diversity extrapolation as a group. For the individual level estimation, animals with <20 sequences were removed. To normalize for sequencing depth, clonal sizes were subsampled 100x to the lowest number of sequences per individual. The average per individual was used as input to estimate Hill numbers, and the Chao1 index was used for plotting[76].

## RABV-G trimer model

A previously published 3D model of trimeric RABV-G (PDB: 7U9G [https://doi.org/10.2210/pdb7U9G/pdb])[77] was downloaded from the protein model database and edited using UCSF ChimeraX (v. 1.2.5). Antigenic sites I, II, and III on the model were identified and highlighted based on previously published epitope mapping[17].

## Competition of the cloned mAbs and plasma with reference rabies antibodies

Competition of the cloned mAbs and plasma with reference rabies antibodies was assessed by ELISA. For the cloned mAbs, an 8-point serial dilution of 1:5 starting from the concentration of 5 µg/mL was used. To provide a reference to calculate competition, an influenza-specific negative control mAb was used. For plasma, samples collected at week 8 were used in a 7-point 4-fold serial dilution. Briefly, mAbs or plasma were added first to RABV-G coated plates and incubated for 30 min. Subsequently, the different reference rabies antibodies, that had been previously biotinylated, were added at fixed concentrations (determined during assay development) and incubated a further 1.5 h. Streptavidin-HRP or Neutravidin-HRP were used for detection. Plates were developed using TMB as described for regular ELISA and absorbance read at 450 nm with 550 nm or 570 nm background correction. For mAbs, competition was assessed by calculating the % reduction of the maximal OD from that of the reference rabies antibodies incubated with the negative controls. For plasma samples, the amount of mAb-competing antibody in each sample was quantified using a standard curve of the respective unbiotinylated mAbs (4-fold 7-point dilution series beginning at 37.5 µg/ml). To evaluate cross-competition between reference mAbs (Supplementary Fig. 12), each reference mAb was run in serial dilution (as for standard curves) against each biotinylated competitor mAb. The area under the curve (AUC) was calculated for each antibody-competitor pairing. The AUC that would result from a non-competing mAb (negAUC) was estimated based on the average of plate blank wells (containing only biotinylated competitor mAb). Cross-competition was estimated using the following formula (here shown for mAb A tested against competitor mAb B): *(negAUC(mAb B)-AUC(mAb A))/(negAUC(mAb B)-AUC(mAb B)\*100%.*

## Statistical methods

Comparisons were made at individual timepoints using Kruskal-Wallis test, with Dunn's test as post-hoc test, adjusting for multiple comparisons using the Benjamini & Hochberg (for BCR sequencing data) or Benjamini, Krieger & Yekutieli methods (all other comparisons) for false-discovery rate when testing differences between three groups. Mann–Whitney $U$ test was used when comparing only two groups. Wilcoxon-signed-rank test was used for paired samples. Two-tailed $p$-value was used in all statistical tests comparing study groups. Spearman correlation with two-tailed $p$-value was used to analyze correlations between different parameters. Non-parametric tests were used as the low sample size did not permit assumption of normal distribution. Statistical analyses were performed using Graphpad Prism version 9.0

(La Jolla California, USA) or R for Mac OS X. A $p$-value < 0.05 was considered as significant for all comparisons.

## Reporting summary

Further information on research design is available in the Nature Portfolio Reporting Summary linked to this article.

## Data availability

The sequencing B cell receptor data generated in this study have been deposited in the NCBI database under a accession code PRJNA932987. The processed B cell receptor data are available at Zenodo repository under the accession code 7680334. Source data are provided with this paper.

## Code availability

The code generated for B cell receptor sequence analysis is publicly available on a Github page (https://rodrigarc.github.io/rabies_mRNA) based on the Github repo (https://github.com/rodrigarc/rabies_mRNA), and at the Zenodo repository under the accession code 7680334.

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

## Acknowledgements

We thank the personnel at the Astrid Fagræus laboratory at Karolinska Institutet for taking care of the animals included in this study. This work was supported by grants INV-005549 and OPP1192908 from the Bill & Melinda Gates Foundation and we thank Holger Kanzler for operating as the senior scientific program officer for this study. We also thank Kansas State University for help with the neutralization assay for the longitudinal samples. The computations were enabled by resources provided by the National Academic Infrastructure for Supercomputing in Sweden (NAISS) and the Swedish National Infrastructure for Computing (SNIC) at UPPMAX partially funded by the Swedish Research Council through grant agreements no. 2022-06725 and no. 2018-05973. The National Bioinformatics Infrastructure Sweden at SciLifeLab provided bioinformatics support.

## Author contributions

Experimental study design: B.P. and K.L. Providing of vaccines and reagents, study logistics: B.P., E.J., and K.S. Acquisition and sample processing: A.C., F.H., A.L., and B.E. Generation of data: A.C., F.H., R.A.C., T.K., S.O., C.F., T.M., and M.C. Analysis and interpretation of data: A.C., F.H., R.A.C., B.P., M.G.D., C.F., T.M., M.C., G.K.H., and K.L. Critical revision of the manuscript: all authors. Statistical analysis: A.C., F.H., R.A.C., M.G.D., and S.O.

## Funding

## Competing interests

E.J., K.S., and B.P. are employees of CureVac SE. The remaining authors declare no competing interests.
