## [Peer Review File · Nature Communications]

Unmodified rabies mRNA vaccine elicits high cross-neutralizing antibody titers and diverse B cell memory responsesReviewers' Comments:

Reviewer #1:

Remarks to the Author:

The manuscript by Cagigi et al investigates the immunogenicity of an mRNA-based vaccine expressing the RABV-G protein in a non-human primate immunisation model. In comparison to licensed vaccine Rabipur, mRNA vaccines induced pro-inflammatory cytokines in the plasma in the 1 day post injection and generally elicited stronger T and B cell immunity following immunisation. A limited analysis of B cell clones and recombinant antibodies was carried out, demonstrating vaccination in NHP could elicit neutralising antibodies with specificities similar to known human antibodies with potent viral neutralising activity. This is a nice study demonstrating the utility of mRNA vaccines to bolster immunity against viral pathogens, and utilising highly translatable NHP models. I have some comments below:

Comments:

1 – While the elicited T cell response is dominated by IFN-g and IL-2, it is worth noting that TH2 and TFH cytokine expression is intrinsically more difficult to detect (evidenced by the SEB controls in Figure S5). This should be caveated when talking about Th1 “polarisation” ie Figure 2h.

2 – The authors compare serum antibodies binding to RABV-G and Rabipur, concluding that “no significant proportion of antibodies to other proteins displayed on the inactivated virus (Supplementary fig. 7a)”. Based on the methodology (ELISA of 100ng/ml RABV-G versus lectin capture Rabipur) I don't see how the authors conclude this using EC50s alone? All the data seems to show is that RABV-G binding antibodies are in proportion to total Rabipur antibodies. They would need to do depletion/absorption experiments, or use recombinant non-G proteins, to establish the extent of antibody response against non-G targets.

3 – “In addition, there was a strong correlation between the titers to Rabipur and RABV-G as well as between Rabipur and neutralization (Supplementary fig. 7b), indicating that the neutralizing antibodies are against RABV-G in both groups.” - while this is likely true, the correlation alone does not prove this and would need to do depletion studies of anti-RABV-G antibodies or similar to mechanistically establish this.

4- “Due to the semi-categorical nature of RVNA titers, we performed the simulation with total RABV-G binding titers instead of RVNA and used EC50 of 80.9 as a target.” – It should be noted that the Rabipur animals did neutralise better than the single shot mRNA animals at peak, and then equivalently over time after that (Fig2a) despite lower antibody binding titres indicating a qualitatively better response (also backed up somewhat by SHM and avidity data). So while the modelled decay rates of binding antibodies indicate a single shot of mRNA is “more durable” than Rabipur (two shot), in terms of neutralisation durability it appears these two are equivalent in the data and this should be flagged/discussed in the text.

5 – “Significantly higher SHM was found in the RABV-G-specific MBC from animals receiving mRNA vaccine once or twice compared to those receiving Rabipur (Figure 4a).” While this might be true when considering the mean/median, the distribution of sequences from the two-shot Rabipur injected animals is distinctly bimodal, with many clones displaying markedly mutation than seen in Group 1 (single shot mRNA). Similarly, the analysis of clonality is highly influenced by sequencing depth. Overall, the number of total sequences sampled in each group is very low and I am not sure any strong conclusions should be drawn on such a limited dataset.

6 – Figure 5a and 5b look basically identical between the groups. Given the low sequencing depth overall, I don't think statements such as “that two mRNA doses boosted a broader IGHV gene usage compared to one mRNA dose” can be well supported. Similarly, estimates of sample coverage and

clonotype diversity seem better left for NGS datasets, and not on pooled sequences from multiple animals. With low sequencing depth these analyses are best excluded as they have a high likelihood of being incorrect and/or misleading. Overall I am not sure Figure 5 adds much knowledge gain to the study overall and I would consider removal.

7 -Figure 5e. This would need some statistical support to allow conclusions such as "mRNA vaccine induced a broader response with antibodies competing for both Site I and Site III".

8 – The Discussion is well written and argued, with potential exception of confounders listed above.

Reviewer #2:

Remarks to the Author:

Let me first apologize for the delay in reviewing this manuscript NCOMMS-22-29707 entitled "Unmodified mRNA vaccine encoding rabies virus glycoprotein induces superior immunity over licensed vaccine in macaques" by Alberto Cagigi et al. This is due to the fact that I remarked, after having accepted the review, that I had previously reviewed an almost identical manuscript submitted to another journal. I mentioned this point to the Editor saying that it could be unfair for the authors (or for the journal) to get twice the same reviewer. I was replied that my expertise in rabies was of interest. This is a good point since I am more a rabies virologist than a fundamental immunologist. Authors will then not be surprised to discover a very similar review than the previous one, the manuscript has almost not evolved since its previous version.

Indeed, the presented data is a very detailed comparative analysis of the immunological responses induced in macaques by vaccination with a nanoparticles-formulated mRNA vaccine coding the RABV-G protein or with one of the classical inactivated vaccines (Rabipur) pre-qualified by WHO. The mRNA vaccine used was formerly developed (ref 22-23). Authors have extensively followed during one year 18 animals in many important steps from innate to adaptive immune responses. Groups of 6 monkeys received either a single dose (100 µg) or two doses at 4 weeks interval of mRNA vaccine while Rabipur vaccine (« full human dose ») was only tested in two doses at 4 weeks interval.

Results globally show that 1 dose of mRNA was comparable to 2 doses of Rabipur, both of them being less performant than 2 doses of mRNA. This is demonstrated at the level (1) of the type-1 IFN response, (2) of the titre and longevity of RABV-G specific Abs (modelized up to 3.5 years) as well as of the B and T cell populations essential to Ab production. In addition it is shown that mRNA vaccination induces qualitative difference in the elicited Abs with more somatic hypermutations (particularly on the antigenic site 1 of the RABV-G protein) while Rabipur vaccination is more oligoclonal, even if no correlation is shown between the degree of hypermutation and the neutralizing potency of RABV-G.

It must be outlined that the amount of results presented as well as the diversity of the methods used are impressive and accumulated by the authors to demonstrate the higher magnitude and quality of the immune responses elicited by the mRNA vaccine over the currently licensed vaccine Rabipur. I am impressed by the number of figures (shown and supp) but I am leaving the Nature Com Editor deciding if s/he feels this detailed analysis can constitute a breakthrough in rabies vaccinology. The remarks/questions thereafter are probably more basic in my (may be simplistic) view of a rabies virologist.

Main points :

- In the M & M as well as in the main text, the origin of the RABV-G protein used for Elisa or binding is not mentioned. This is however of importance since there are substantial genetic differences between the rabies vaccine strains (see fig 2 of <https://doi.org/10.1099/0022-1317-73-5-1149>). Rabipur is

using the strain Flury LEP which may be different from the strain expressed by the authors (in which expression/purification system ?) and which is different from the CVS strain orthodoxy used for the evaluation of rabies neutralizing antibodies as claimed to be performed by the authors. Finally, the ARNm is that of the Pasteur virus strain which is indeed from South-American origin. As authors are looking to subtle differences between tests, this is very important to consider.

- Results I. 171-3 : « There was no difference in titers to Rabipur compared to RABV-G in the Rabipur group, demonstrating no significant proportion of antibodies to other proteins displayed on the inactivated virus (Fig. S7A) ». Why observing the same titre for binding to G protein between Rabipur and mRNA would interfere with the recognition of other antigens present in Rabipur ? Have other proteins (N, other) been tested ? BTW this is a clear point which may differ an inactivated vaccine, which is exposing all proteins to the immune system, from a mRNA vaccine specific for G. If the manuscript is logically focusing the response to the G protein, a non-tested response to other viral antigens could possibly help the global response in case of challenge with a complete virus (cross-help or so).

- As a naïve immunologist, I am questioning myself about the interest of having a polyclonal response (mRNA) versus a focused oligoclonal one (Rabipur). Particularly as (I. 251-2) “the neutralizing potency of RABV-G binding antibodies was thus found to be independent of the degree of SHM (fig. 4G-F)” . To enlarge this remark, one could be surprised that most of the Abs generated by the vaccination and further cloned were focusing sites I and III. Indeed, the reference 12 cited in the manuscript indicated that about 75% of the generated mouse Mabs were recognising the antigenic site II which is separated in two epitopes put into contact by the 3D folding of the G protein. It is surprising not to observe any of them in those selected in the manuscript, unless the screening would have been done against a defolded G protein.

- In the same line, (I. 283-4) authors claim that the mRNA vaccine induces a broader response against site I and III than the Rabipur.

- o Yes but they seem to compete less, meaning that they are more dispersed but not systematically more inhibitory

- o It would have been interesting to confirm this assumption by testing the sera in vitro against different viruses in the species rabies or even against lyssaviruses from different species (binding or VNabs).

- The ultimate demonstration would have been the challenge of the monkeys but I perfectly understand that it is going behind to the present manuscript. During challenge, one can imagine that other viral proteins present in Rabipur could play a role. Just for memory, while RNA vaccine has received a very solid boost from the COVID epidemics, we have to remember that as soon as 2000, dogs were fully protected from a peripheral challenge after injection of 100µg of plasmid DNA (same amount than here) encoding the PV-strain RABV-G protein ... (Perrin et al. (2000) Immunization of dogs with a DNA vaccine induces protection against rabies virus Vaccine 18, 479-486).

Other points :

- Title : « Unmodified mRNA vaccine encoding rabies virus glycoprotein induces superior immunity over licensed vaccine in macaques » is a bit abusive since the manuscript is only considering one licensed vaccine (Rabipur) while others exist using different vaccine strains as outlined earlier (for example, Sanofi-Pasteur is using the PM strain, some American vaccines are using the PV or the ERA strains, etc). For that reason, it would be more coherent to conclude the title by « ... over one (or a) licensed vaccine in macaques »

- Abstract : For reasons mention above the sentence I.32-33 should be tempered in : “However, the mRNA vaccine induced higher degree of somatic hypermutation, polyclonality and targeting of more antigenic sites ON THE G PROTEIN »

- Introduction, l. 59: An important reference could be cited l.296, together with ref 27 : "Hampson, K.; Coudeville, L.; Lembo, T.; Sambo, M.; Kieffer, A.; Attlan, M.; Barrat, J.; Blanton, J.D.; Briggs, D.J.; Cleaveland, S et al. Estimating the global burden of endemic canine rabies. PLoS Negl. Trop. Dis. 2015, 9, e0003709.
- Results l. 105-7 "The mRNA vaccine induced significant increases in serum concentrations of interferon alpha (IFN α) as well as IFN-inducible I-TAC/CXCL11, which were not detectable in the Rabipur group (Fig. 1B) ». Indeed they are poorly and transiently detectable.
- L. 218 : not clear to me if figure 4B is the good one
- L.454-5 « This study was approved by the Local Ethical Committee on Animal Experiments ». Is it a reference, a number ?

Reviewer #3:

Remarks to the Author:

In this manuscript, the authors showed that two doses of a lipid nanoparticle-formulated unmodified mRNA vaccine encoding the rabies virus glycoprotein (RABV-G) induced higher levels of RABV-G specific plasmablasts, memory B cells and T cells in blood, and plasma cells in the bone marrow compared to two doses of Rabipur in non-human primates. The authors claimed that the mRNA vaccine induced higher degree of somatic hypermutation, polyclonality and targeting of more antigenic sites. The manuscript is well-written and the data are well-presented. However, some ideas are not well supported by the experiments. For the benefit of the reader, some concerns should be addressed:

Major points:

1. It is interesting that Rabipur vaccine showed more oligoclonal profile (induce more BCR clonal expansion) than the mRNA prime-boost group (Fig 4B). Please explain it. Moreover, are the number of BCR clones greater than 2 specify against the RABV-G protein? Can the author isolated these monoclonal antibodies and characterize them.
2. The authors have sorted 334 single MBC. Only 12 pairs of heavy and light chain mAbs were purified. More mAbs from group 2 and 3 should be isolated and characterized to supported the idea "mRNA vaccination generated a broader response with more antibodies directed towards Site I compared to Rabipur".

Mini points:

1. The unit ug/ml should be changed to μ g/ml.
2. The format of the references should be checked carefully.

Dear reviewers,

Thank you for the constructive criticism on our manuscript no NCOMMS-22-2970 entitled **“Unmodified mRNA vaccine encoding rabies virus glycoprotein induces superior immunity compared to a licensed vaccine in macaques”**. We are grateful for the opportunity to revise our work and apologize for the long revision time. This is due to that we have put considerable efforts into performing additional experiments as requested by the reviewers. We thank you for your insightful comments, which greatly helped us improve this study to become more compelling.

Below is a detailed list of how the manuscript has been changed in the response to each of the comments. Other changes of the manuscript were made in response to comments added by the new authors and to adjust the length.

REVIEWER COMMENTS

Reviewer #1 (Remarks to the Author):

The manuscript by Cagigi et al investigates the immunogenicity of an mRNA-based vaccine expressing the RABV-G protein in a non-human primate immunisation model. In comparison to licensed vaccine Rabipur, mRNA vaccines induced pro-inflammatory cytokines in the plasma in the 1 day post injection and generally elicited stronger T and B cell immunity following immunisation. A limited analysis of B cell clones and recombinant antibodies was carried out, demonstrating vaccination in NHP could elicit neutralising antibodies with specificities similar to known human antibodies with potent viral neutralising activity. This is a nice study demonstrating the utility of mRNA vaccines to bolster immunity against viral pathogens, and utilising highly translatable NHP models. I have some comments below:

Comments:

1 – While the elicited T cell response is dominated by IFN-g and IL-2, it is worth noting that TH2 and TFH cytokine expression is intrinsically more difficult to detect (evidenced by the SEB controls in Figure S5). This should be caveated when talking about Th1 “polarisation” ie Figure 2h.

A: We agree with the reviewer. This has now been stated in the manuscript (Page 4 in the revised manuscript).

2 – The authors compare serum antibodies binding to RABV-G and Rabipur, concluding that “no significant proportion of antibodies to other proteins displayed on the inactivated virus (Supplementary fig. 7a)”. Based on the methodology (ELISA of 100ng/ml RABV-G versus lectin capture Rabipur) I don’t see how the authors conclude this using EC50s alone? All the data seems to show is that RABV-G binding antibodies are in proportion to total Rabipur antibodies. They would need to do depletion/absorption experiments, or use recombinant non-G proteins, to establish the extent of antibody response against non-G targets.

A: This point is well taken. We have now performed ELISA for antibodies against the RABV-Nucleoprotein (N) in order to analyze another key antigen to which antibodies have been demonstrated an effect in rabies challenge studies in mice (Lodmell et al, PMID:8371354). However, we found undetectable or very low levels of anti-N antibodies in the Rabipur immunized animals. The new data have been added to Supplementary figure 7c. While these data alone do not demonstrate that there are only responses against RABV-G induced by Rabipur immunization, it seems likely that the vast majority of the protective response is directed to this protein. We have toned down our interpretation and rephrased the text in the revised manuscript.

3 – “In addition, there was a strong correlation between the titers to Rabipur and RABV-G as well as

between Rabipur and neutralization (Supplementary fig. 7b), indicating that the neutralizing antibodies are against RABV-G in both groups.” - while this is likely true, the correlation alone does not prove this and would need to do depletion studies of anti-RABV-G antibodies or similar to mechanistically establish this.

A: As discussed above, we agree with the reviewer that a correlation alone is not conclusive. Antibodies to other viral proteins may contribute to the overall response although the majority of neutralization is likely mediated by RABV-G-specific antibodies as reposted (Dietzschold et al, PMID 3672933). It has been shown for other whole inactivated virus vaccines e.g. influenza and SARS-CoV-2 that most of the antibody response is directed to the protein(s) expressed on the viral surface. T cell responses may be generated to internal proteins since they are processed and presented during antigen presentation. However, T cell responses may not play a significant role in the protection of rabies virus infection. We were unable to perform depletion studies of anti-G antibodies but have clarified uncertainties with the interpretation and that such analyses are required to definitively show if there is any effect of antibodies specific to other proteins.

4- *“Due to the semi-categorical nature of RVNA titers, we performed the simulation with total RABV-G binding titers instead of RVNA and used EC50 of 80.9 as a target.” – It should be noted that the Rabipur animals did neutralise better than the single shot mRNA animals at peak, and then equivalently over time after that (Fig2a) despite lower antibody binding titres indicating a qualitatively better response (also backed up somewhat by SHM and avidity data). So while the modelled decay rates of binding antibodies indicate a single shot of mRNA is “more durable” than Rabipur (two shot), in terms of neutralisation durability it appears these two are equivalent in the data and this should be flagged/discussed in the text.*

A: We thank the reviewer for bringing this excellent point to our attention. We have now highlighted this accordingly in the text (Page 4-5 in the revised manuscript). We have also now commented on in the discussion that this may be supported by that slightly lower RABV-G binding titers showed a bit more potent neutralization in this group (Figure 3a).

5 – *“Significantly higher SHM was found in the RABV-G-specific MBC from animals receiving mRNA vaccine once or twice compared to those receiving Rabipur (Figure 4a).” While this might be true when considering the mean/median, the distribution of sequences from the two-shot Rabipur injected animals is distinctly bimodal, with many clones displaying markedly mutation than seen in Group 1 (single shot mRNA). Similarly, the analysis of clonality is highly influenced by sequencing depth. Overall, the number of total sequences sampled in each group is very low and I am not sure any strong conclusions should be drawn on such a limited dataset.*

A: We understand that the sequencing depth in the original version of the manuscript was low and a weakness of the study. We have therefore put considerable efforts into strengthening this part to be able to solidify the results. We have now sorted several more RABV-G specific memory B cells and sequenced in total 410 and 506 cells from the mRNA vaccine and Rabipur prime-boost groups, respectively. In addition, we performed bulk IgG heavy chain VDJ repertoire sequencing from all immunized animals at week 18, from which we searched for clonally related RABV-G specific sequences using the data from single cell sequencing as query sequences. We utilized this more extensive dataset to re-analyze SHM and clonotype diversity. Figure 4 and 5 have been updated and rearranged with the new data.

With regards to SHM, by including the new sequences from the larger number of sorted RABV-G specific memory B cells, we found that SHM in the prime-boosted mRNA vaccine group was overall higher than in the prime-boosted Rabipur group. However, this was not significantly different on an individual animal level. We have therefore clarified this and rephrased the conclusion. The SHM data on an individual animal level are shown in the new Figure 4a.

6 – *Figure 5a and 5b look basically identical between the groups. Given the low sequencing depth overall, I*

don't think statements such as "that two mRNA doses boosted a broader IGHV gene usage compared to one mRNA dose" can be well supported. Similarly, estimates of sample coverage and clonotype diversity seem better left for NGS datasets, and not on pooled sequences from multiple animals. With low sequencing depth these analyses are best excluded as they have a high likelihood of being incorrect and/or misleading. Overall I am not sure Figure 5 adds much knowledge gain to the study overall and I would consider removal.

A: As mentioned above, we have now increased the sequencing depth by analyzing more RABV-G specific memory B cells as well as bulk IgG repertoire sequencing data, which generated 12.5 million sequences from the 12 animals in the mRNA and Rabipur prime-boosted groups. In brief, clonally related heavy chain sequences obtained from the RABV-G specific memory B cells were searched for in the large donor-matched IgG repertoire dataset (clonotype definition: matching IGHV and IGJH allele assignment, unchanged HCDR3 length, and 80% HCDR3 amino acid homology), which added considerably to the sequence depth. We focused on the comparison between the groups receiving two doses of vaccines. With regards to clonal diversity, we updated Figure 5 with the new data and, for full transparency, this figure now illustrates the distribution and number of sequences obtained per animal and group. With this data, we concluded that although there is a trend that mRNA vaccination induced a more polyclonal response as compared to Rabipur, there is donor variability in both groups and not a clear difference between the groups. We feel that with this more extensive approach the data are likely more correct and we conclude that there is no significant difference for these read-outs between the two vaccines. However, as we describe below, cross-neutralization to other virus strains was significantly better by mRNA vaccination. Our interpretation is that the higher frequencies of activated B cells and the higher antibody titers induced in the mRNA vaccine group are the main reasons for the increased breadth and cross-neutralization observed in that group. We have therefore rewritten the text both in the result and discussion sections. We thank the reviewer for asking us to do further analyses to strengthen our analyses.

7 -Figure 5e. This would need some statistical support to allow conclusions such as "mRNA vaccine induced a broader response with antibodies competing for both Site I and Site III".

A: The data in figure 5e has been updated with Site II competition. We also now display the competition data in another format, such that comparisons between the groups are easier to see and statistical calculations facilitated.

8 – The Discussion is well written and argued, with potential exception of confounders listed above.

A: Thanks for the comment. We have re-written parts of the discussion to reflect the new data and made sure to bring up both potential benefits but also risks and disadvantages with the mRNA vaccine technology. We have emphasized the significant value of the existing licensed whole inactivated virus vaccines such as Rabipur have had in saving numerous lives is highlighted.

Reviewer #2 (Remarks to the Author):

Let me first apologize for the delay in reviewing this manuscript NCOMMS-22-29707 entitled "Unmodified mRNA vaccine encoding rabies virus glycoprotein induces superior immunity over licensed vaccine in macaques" by Alberto Cagigi et al. This is due to the fact that I remarked, after having accepted the review, that I had previously reviewed an almost identical manuscript submitted to another journal. I mentioned this point to the Editor saying that it could be unfair for the authors (or for the journal) to get twice the same reviewer. I was replied that my expertise in rabies was of interest. This is a good point since I am more a rabies virologist than a fundamental immunologist. Authors will then not be surprised to discover a very similar review than the previous one, the manuscript has almost not evolved since its previous version.

Indeed, the presented data is a very detailed comparative analysis of the immunological responses induced in macaques by vaccination with a nanoparticles-formulated mRNA vaccine coding the RABV-G protein or with one of the classical inactivated vaccines (Rabipur) pre-qualified by WHO. The mRNA vaccine used was formerly developed (ref 22-23). Authors have extensively followed during one year 18 animals in many important steps from innate to adaptive immune responses. Groups of 6 monkeys received either a single dose (100 µg) or two doses at 4 weeks interval of mRNA vaccine while Rabipur vaccine (« full human dose ») was only tested in two doses at 4 weeks interval.

Results globally show that 1 dose of mRNA was comparable to 2 doses of Rabipur, both of them being less performant than 2 doses of mRNA. This is demonstrated at the level (1) of the type-1 IFN response, (2) of the titre and longevity of RABV-G specific Abs (modelized up to 3.5 years) as well as of the B and T cell populations essential to Ab production. In addition it is shown that mRNA vaccination induces qualitative difference in the elicited Abs with more somatic hypermutations (particularly on the antigenic site 1 of the RABV-G protein) while Rabipur vaccination is more oligoclonal, even if no correlation is shown between the degree of hypermutation and the neutralizing potency of RABV-G.

It must be outlined that the amount of results presented as well as the diversity of the methods used are impressive and accumulated by the authors to demonstrate the higher magnitude and quality of the immune responses elicited by the mRNA vaccine over the currently licensed vaccine Rabipur. I am impressed by the number of figures (shown and supp) but I am leaving the Nature Com Editor deciding if s/he feels this detailed analysis can constitute a breakthrough in rabies vaccinology. The remarks/questions thereafter are probably more basic in my (may be simplistic) view of a rabies virologist.

A: We thank the reviewer for the positive feedback and the nice description of our study. We second the editor that an in-depth expertise on rabies will help us complete and strengthen our study, for which we have now teamed up with a rabies laboratory to be able to conduct more experiments and modified the manuscript substantially according to the reviewer's comments below.

Main points :

- In the M & M as well as in the main text, the origin of the RABV-G protein used for Elisa or binding is not mentioned. This is however of importance since there are substantial genetic differences between the rabies vaccine strains (see fig 2 of <https://doi.org/10.1099/0022-1317-73-5-1149>). Rabipur is using the strain Flury LEP which may be different from the strain expressed by the authors (in which expression/purification system ?) and which is different from the CVS strain orthodoxy used for the evaluation of rabies neutralizing antibodies as claimed to be performed by the authors. Finally, the ARNm is that of the Pasteur virus strain which is indeed from South-American origin. As authors are looking to subtle differences between tests, this is very important to consider.

A: We apologize that this was not clear in the manuscript. This information has been added to the revised version. The mRNA vaccine is encoding for the Pasteur virus strain which is also the strain of the recombinant RABV-G protein used in our immunological assays (ELISA, B cell probing). We have now also added information of the expression system. The strain used for VNT testing was the CVS strain as correctly pointed out by the reviewer. As suggested below by the reviewer we have now also performed neutralization measurements against additional lyssaviruses from different species (described more below).

- Results I. 171-3 : « There was no difference in titers to Rabipur compared to RABV-G in the Rabipur group, demonstrating no significant proportion of antibodies to other proteins displayed on the inactivated virus (Fig. S7A) ». Why observing the same titre for binding to G protein between Rabipur and mRNA would interfere with the recognition of other antigens present in Rabipur ? Have other proteins (N, other) been tested ? BTW this is a clear point which may differ an inactivated vaccine, which is exposing all

proteins to the immune system, from a mRNA vaccine specific for G. If the manuscript is logically focusing the response to the G protein, a non-tested response to other viral antigens could possibly help the global response in case of challenge with a complete virus (cross-help or so).

A: As mentioned in the response to Reviewer 1, comment 2; This point is well taken. We have now performed ELISA for antibodies against RABV-Nucleoprotein (N) as another key antigen to which antibodies have been demonstrated a protective effect in rabies challenge studies in mice (Lodmell et al, PMID: 8371354). However, we found low or undetectable levels of anti-N antibodies in the Rabipur group. The new data have been added to Supplementary figure 7c. While this data is still not demonstrating that there are only responses against RABV-G induced by Rabipur immunization, it seems likely that the vast majority of the neutralizing response is directed to this protein. We have toned down our interpretation and rephrased the text in the revised manuscript. We agree that a response to other viral proteins may contribute to the global response. It has been shown for other whole inactivated virus vaccines e.g., influenza and SARS-CoV-2 that most of the antibody response is directed to the proteins expressed on the viral surface. T cell responses may be generated to internal proteins since they are processed for antigen presentation. However, T cell responses may not play a significant role in the protection or control of rabies virus infection.

• As a naïve immunologist, I am questioning myself about the interest of having a polyclonal response (mRNA) versus a focused oligoclonal one (Rabipur). Particularly as (l. 251-2) “the neutralizing potency of RABV-G binding antibodies was thus found to be independent of the degree of SHM (fig. 4G-F)” . To enlarge this remark, one could be surprised that most of the Abs generated by the vaccination and further cloned were focusing sites I and III. Indeed, the reference 12 cited in the manuscript indicated that about 75% of the generated mouse Mabs were recognising the antigenic site II which is separated in two epitopes put into contact by the 3D folding of the G protein. It is surprising not to observe any of them in those selected in the manuscript, unless the screening would have been done against a defolded G protein.

A: We realize that it was not clear in the manuscript that we had intentionally focused on site I and III motivated by that they are the binding sites for several licensed therapeutic antibodies plus for highly potent antibodies identified in screening efforts. We have now performed competition assays with reference Site II specific antibodies similar to what we have done with Site I and III specific antibodies. We observed that there are indeed antibody responses induced to Site II. This was found both by the mRNA vaccine as well as by Rabipur. The new data have been added to figure 5e. We found that the animals that received the mRNA vaccine showed superior competition with all reference mAbs except Rafivirumab compared to the response induced by Rabipur. This demonstrates that some level of response to antigenic Sites I, II and III on RABV-G is elicited by both vaccines but this is more restricted by Rabipur likely due to the lower titers. The main advantage with the mRNA vaccine is the stronger induction of immunity. We have replaced the word “superior” to “stronger” in the title as it felt more accurate.

*• In the same line, (l. 283-4) authors claim that the mRNA vaccine induces a broader response against site I and III than the Rabipur.
o Yes but they seem to compete less, meaning that they are more dispersed but not systematically more inhibitory*

A: Figure 5e has now been updated with Site II competition. We also now display the data in another format such that comparisons between the groups are easier to see and statistical calculations facilitated.

o It would have been interesting to confirm this assumption by testing the sera in vitro against different viruses in the species rabies or even against lyssaviruses from different species (binding or Vnabs).

A: We have now expanded the VNT analyses of neutralization capacity against lyssaviruses from different species induced after mRNA vaccine vs Rabipur with the help of new collaborators. Neutralization capacity at week 47 from all the animals was tested. The mRNA vaccinated animals showed cross-neutralizing responses to EBLV1 and DUVV, while this was lower in the Rabipur immunized animals. No neutralization to LBV or LLBEV was found in either group. This again suggests that the improved responses found with the mRNA vaccine are largely due to the higher antibody levels leading to increased diversity and breadth. The new data are shown in Figure 5f.

• *The ultimate demonstration would have been the challenge of the monkeys but I perfectly understand that it is going behind to the present manuscript. During challenge, one can imagine that other viral proteins present in Rabipur could play a role. Just for memory, while RNA vaccine has received a very solid boost from the COVID epidemics, we have to remember that as soon as 2000, dogs were fully protected from a peripheral challenge after injection of 100µg of plasmid DNA (same amount than here) encoding the PV-strain RABV-G protein ... (Perrin et al. (2000) Immunization of dogs with a DNA vaccine induces protection against rabies virus Vaccine 18, 479-486).*

A: We were unable to challenge the monkeys but for the interest of the reviewer we attach the results here from an experiment where mice were challenged after receiving the same mRNA vaccine (1/20 of the dose used in the monkeys) or Rabipur (1/10 of the dose). This showed that both the mRNA vaccine at one or two doses and Rabipur at two doses gave full protection.

Figure legend: **A:** Neutralizing antibody titers after vaccination, prior to challenge. Dotted line indicates WHO recommended threshold of 0.5 IU/ml. Triangles indicate immunizations. Six weeks week old NIH Swiss mice (purchased from Envigo) were injected intramuscularly (IM) twice 28 days apart with one or two doses of 5 µg rabies G mRNA-LNP vaccine (CV7202). The positive control group received one tenth of the human dose of the licensed inactivated rabies vaccine Rabipur, three times on day 0, 7 and 28 via the IM route. Mice in the negative control group received physiological saline (0.9% NaCl) twice 28 days apart via the IM route. Three weeks after the last vaccination the mice were challenged with $10^{2.2}$ TCID₅₀ of CVS-11 rabies virus via the intracranial route. **B:** Percentage body weight loss following intracranial virus challenge three weeks after final immunization. **C:** Survival following intracranial virus challenge. Following the rabies challenge infection, the animals were monitored for up to 21 days including daily measurement of body weight and assessment of clinical status using a scoring system (phase 1: ruffled hair as early sign of encephalitis, phase 2: ruffled hair /

hunched back as sign of progression to rabies encephalitis, phase 3: ruffled hair / hunched back / hind leg(s) paralysis as sign of rabies encephalitis). Animals were euthanized upon reaching phase 3 (humane end point for euthanasia). Additionally, mice that lost body weight on two following days of > 5% were euthanized, as based on previous experiments, the animals will progress to rabies encephalitis.

We also thank the reviewer for pointing to the work of Perrin et al. This is a good reference which has now been added to the manuscript in the discussion.

Other points :

- *Title : « Unmodified mRNA vaccine encoding rabies virus glycoprotein induces superior immunity over licensed vaccine in macaques » is a bit abusive since the manuscript is only considering one licensed vaccine (Rabipur) while others exist using different vaccine strains as outlined earlier (for example, Sanofi-Pasteur is using the PM strain, some American vaccines are using the PV or the ERA strains, etc). For that reason, it would be more coherent to conclude the title by « ... over one (or a) licensed vaccine in macaques »*

A: We agree with the reviewer and the title has now been changed accordingly.

- *Abstract : For reasons mention above the sentence l.32-33 should be tempered in : “However, the mRNA vaccine induced higher degree of somatic hypermutation, polyclonality and targeting of more antigenic sites ON THE G PROTEIN »*

A: The abstract has now been revised to reflect the new data and this sentence has been revised.

- *Introduction, l. 59: An important reference could be cited l.296, together with ref 27 : “Hampson, K.; Coudeville, L.; Lembo, T.; Sambo, M.; Kieffer, A.; Attlan, M.; Barrat, J.; Blanton, J.D.; Briggs, D.J.; Cleaveland, S et al. Estimating the global burden of endemic canine rabies. PLoS Negl. Trop. Dis. 2015, 9, e0003709.*

A: We thank the reviewer for valuable input. This reference has now been added.

- *Results l. 105-7 “The mRNA vaccine induced significant increases in serum concentrations of interferon alpha (IFN α) as well as IFN-inducible I-TAC/CXCL11, which were not detectable in the Rabipur group (Fig. 1B) ». Indeed they are poorly and transiently detectable.*

A: This has now been modified in the text.

- *L. 218 : not clear to me if figure 4B is the good one*

A: We agree with the reviewer that the previous version of Figure 4b could be made more informative. Figure 4 has now been rearranged and updated with the much larger data set generated by sorting and sequencing more RABV-G specific memory B cells as well as bulk Ig repertoire sequencing. Diversity in terms of clonal expansion is now shown in Figure 5b and is now presented as a bar graph also illustrating the distribution and number of sequences obtained per animal and group for transparency.

- *L.454-5 « This study was approved by the Local Ethical Committee on Animal Experiments ». Is it a reference, a number ?*

A: The approval number has been added.

Reviewer #3 (Remarks to the Author):

In this manuscript, the authors showed that two doses of a lipid nanoparticle-formulated unmodified mRNA vaccine encoding the rabies virus glycoprotein (RABV-G) induced higher levels of RABV-G specific plasmablasts, memory B cells and T cells in blood, and plasma cells in the bone marrow compared to two doses of Rabipur in non-human primates. The authors claimed that the mRNA vaccine induced higher degree of somatic hypermutation, polyclonality and targeting of more antigenic sites. The manuscript is well-written and the data are well-presented. However, some ideas are not well supported by the experiments. For the benefit of the reader, some concerns should be addressed:

Major points:

1. It is interesting that Rabipur vaccine showed more oligoclonal profile (induce more BCR clonal expansion) than the mRNA prime-boost group (Fig 4B). Please explain it.

A: It has been demonstrated in a mouse model that membrane-bound antigen from mRNA vaccination was better than a protein vaccine at activating B cells by lowering the affinity threshold needed for recruitment of B cells into germinal centers (Melzi et al, PMID: 36179690). An array of proteins expressed on the surface of a cell after mRNA vaccine uptake may therefore be a strong stimuli for B cell activation and the recruitment of B cells with a wider range of initial affinities. However, more studies are needed to understand this. The chemical inactivation of virus with β -propiolactone as used for Rabipur can alter antigen properties and potentially destroy certain epitopes as found for other virus preparations and this could result in narrower antigen specificities developed. However, the work we performed during this revision revealed that the clonal diversity may not be that different between the mRNA vaccine and Rabipur as we originally had thought. The text has been updated to reflect the new data. Nevertheless, the mRNA vaccinated animals in our study showed higher frequencies of B cells, plasma cells, T cells and antibody titers as well as cross-neutralization so there are clearly advantages with induction of strong immunity.

Moreover, are the number of BCR clones greater than 2 specify against the RABV-G protein? Can the author isolated these monoclonal antibodies and characterize them.

A: We sorted memory B cells by using RABV-G probes conjugated to two different fluorochromes. We sorted single B cells that were binding both probes to reduce unspecific binders. When we selected sequences to express monoclonal antibodies, we considered several factors such as sequence quality, matched heavy and light chain, degrees of SHM (low, medium and high) and sequence similarity with well-characterized reference antibodies against Sites I and III on RABV-G. We did therefore not focus on expressing antibodies from expanded memory B cell clones in particular. This would certainly be interesting but since the responses are highly polyclonal and donor-specific, expanded clones are likely different in different animals. Studying whether any specific clone appears in several animals and the expansion of such will be a different study.

2. The authors have sorted 334 single MBC. Only 12 pairs of heavy and light chain mAbs were purified. More mAbs from group 2 and 3 should be isolated and characterized to supported the idea "mRNA vaccination generated a broader response with more antibodies directed towards Site I compared to Rabipur".

A: As also pointed out by the other reviewers, the sequencing depth in the original version of the manuscript was low and a weakness of the study. We have therefore put considerable efforts into strengthening this part to be able to solidify the results. We have now sorted several more RABV-G specific memory B cells and sequenced in total 410 and 506 cells from the mRNA vaccine and Rabipur prime-boost groups, respectively. In addition, we have performed bulk IgG repertoire sequencing from all prime-boost immunized animals, generating around 12.5 million IgG reads from which we searched for related RABV-G specific clonotype sequences. We then utilized this much more

extensive dataset to re-analyze SHM and clonotype diversity. Figure 4 and 5 have been updated with the new data. We have re-written the sections regarding this to reflect the new findings. We feel more confident with the additional experiments that our findings are correct. We agree that cloning even more antibodies would be interesting. However, this would be a large task to take on and we feel that it was a better strategy to embark on generating the new larger sequence dataset first. It would certainly be interesting to use some of the new data to express more antibodies in a subsequent study.

Mini points:

1. The unit ug/ml should be changed to $\mu\text{g/ml}$.
2. The format of the references should be checked carefully.

A: Thank you. We have now corrected these errors.

We believe that the manuscript has been much improved by the additional experiments and edits made to the text. We again thank the reviewers for their insightful comments and for pushing us in the right direction. Several new authors were added in performing these revisions, which is reflected in the edits made to the text. All authors have read and agreed with submission of the revised manuscript.

Kind Regards,

Karin Loré

Reviewers' Comments:

Reviewer #1:

Remarks to the Author:

The authors have effectively and comprehensively addressed my prior concerns (and those of other reviewers). This is a great study, and the manuscript text and results are in better harmony now. I recommend acceptance.

Reviewer #3:

Remarks to the Author:

The authors have addressed all my concerns.

Reviewer #4:

Remarks to the Author:

All answers to comments and suggestions are satisfactory to this reviewer. The added data and adjusted text are much appreciated and solidify the study substantially.

As the last 2 comments here, I would encourage the authors:

- 1- to de-emphasize that the lack of strong adverse events in NHPs (behavioral, food intake etc) as a sign of safety and tolerability in humans. NHPs are overall extremely resistant to vaccine adverse events, and aside from intra-cranial administration, very little can be inferred (maybe that volunteers will likely not die from a low dose of a dose escalating study).
- 2- Mention in the limitations, long-term protection as been a challenge with mRNA vaccines and breakthrough infections well documented. Breakthrough infections with Rabies would likely be fatal in most of the cases (if not treated rapidly).

All of these points (safety and long-term protection / breakthrough infection) will need to be carefully studied and documented before this approach can be widely used.

Final revisions for Nature Communications manuscript NCOMMS-22-29707A – response to reviewer comments

REVIEWERS' COMMENTS

Reviewer #1 (Remarks to the Author):

The authors have effectively and comprehensively addressed my prior concerns (and those of other reviewers). This is a great study, and the manuscript text and results are in better harmony now. I recommend acceptance.

Reviewer #3 (Remarks to the Author):

The authors have addressed all my concerns.

Reviewer #4 (Remarks to the Author):

All answers to comments and suggestions are satisfactory to this reviewer. The added data and adjusted text are much appreciated and solidify the study substantially.

As the last 2 comments here, I would encourage the authors:

1- to de-emphasize that the lack of strong adverse events in NHPs (behavioral, food intake etc) as a sign of safety and tolerability in humans. NHPs are overall extremely resistant to vaccine adverse events, and aside from intra-cranial administration, very little can be inferred (maybe that volunteers will likely not die from a low dose of a dose escalating study).

We have now highlighted in the discussion the lack of vaccine adverse events in NHPs as a caveat and point of consideration when interpreting reactogenicity and safety data from NHP vaccination studies.

2- Mention in the limitations, long-term protection as been a challenge with mRNA vaccines and breakthrough infections well documented. Breakthrough infections with Rabies would likely be fatal in most of the cases (if not treated rapidly).

All of these points (safety and long-term protection / breakthrough infection) will need to be carefully studied and documented before this approach can be widely used.

We have now mentioned breakthrough infections as evident during the SARS-CoV-2 pandemic as a potential issue with waning immunity and highlighted the importance of monitoring durability in the discussion. In our study, however, mRNA vaccinated animals maintained higher antibody titers at study end compared to animals receiving the licensed whole inactivated virus vaccine. Nevertheless, as the reviewer points out breakthrough infections with rabies virus would be fatal and therefore maintenance of antibody levels must be carefully evaluated.